

# Improved rapid landslide detection from integration of empirical models and satellite radar

Katy Burrows[1,4], David Milledge[2], Richard J. Walters[1], and Dino Bellugi[3]

[1]COMET, Department of Earth Sciences, Durham University, Durham, U.K.
[2]School of Engineering, Newcastle University, Newcastle, U.K.
[3]Department of Geography, University of California, Berkeley, U.S.A.
[4]Now at Géoscience Environnement Toulouse, CNES, Toulouse, France

**Correspondence:** Katy Burrows (katy.burrows@get.omp.eu)

**Abstract.** Information on the spatial distribution of triggered landslides following an earthquake is invaluable to emergency responders. Manual mapping using optical satellite imagery, which is currently the most common method of generating this landslide information, is extremely time consuming and can be disrupted by cloud-cover. Empirical models of landslide probability and landslide detection with satellite radar data are two alternative methods of generating information on triggered landslides that overcome these limitations. Here we assess the potential of a combined approach, in which we generate an empirical model of the landslides using data available immediately following the earthquake using the Random Forests technique, and then progressively add landslide indicators derived from Sentinel-1 and ALOS-2 satellite radar data to this model in the order they were acquired following the earthquake. We use three large case study earthquakes and test two model types: first, a model that is trained on a small part of the study area and used to predict the remainder of the landslides, and second a preliminary "global" model that is trained on the landslide data from two earthquakes and used to predict the third. We assess model performance using receiver operating characteristic analysis and r$^2$, and find that the addition of the radar data can considerably improve model performance and robustness within two weeks of the earthquake. In particular, we observed a large improvement in model performance when the first ALOS-2 image was added and recommend that these data or similar data from other L-band radar satellites be routinely incorporated in future empirical models.

## 1  Introduction

Earthquake-triggered landslides are a major secondary hazard associated with large continental earthquakes and disrupt emergency response efforts. Information on their spatial distribution is required to inform this emergency response, but must be generated within two weeks of the earthquake in order to be most useful (Inter-Agency Standing Committee, 2015; Williams et al., 2018). The most common method of generating landslide information is manual mapping using optical satellite imagery,





but this is a time-consuming process and can be delayed by weeks or even months due to cloud cover (Robinson et al., 2019), leading to incomplete landslide information during the emergency response.

In the absence of optical satellite imagery, there are two options for generating information on the intensity and spatial extent of landslides in the immediate aftermath of a large earthquake. The first is to produce empirical susceptibility maps, using factors such as slope, lithology and estimations of ground shaking intensity to predict areas where landslides are likely to have occurred (e.g. Nowicki Jessee et al., 2018; Robinson et al., 2017; Tanyas et al., 2019). The second is to estimate landslide locations based on their signal in satellite synthetic aperture radar (SAR) data, which can be acquired through cloud cover and so is often able to provide more complete spatial coverage than optical satellite imagery in the critical two week response window (e.g. Aimaiti et al., 2019; Burrows et al., 2019, 2020; Jung and Yun, 2019; Konishi and Suga, 2019; Mondini et al., 2019).

To generate an empirical landslide susceptibility model, a training dataset of mapped landslides is analysed alongside maps of factors known to influence landslide likelihood such as slope, land cover and ground shaking estimates, and a model is produced that predicts landslide likelihood based on these inputs. A range of methods have been used to generate landslide susceptibility models, including: Fuzzy Logic (Kirschbaum and Stanley, 2018; Kritikos et al., 2015; Robinson et al., 2017), Logistic Regression (Cui et al., 2020; Nowicki Jessee et al., 2018; Tanyas et al., 2019) and Random Forests (Catani et al., 2013; Chen et al., 2017; Fan et al., 2020). When generating a susceptibility map for emergency response, the training dataset can be either a collection of landslide inventories triggered by multiple earthquakes worldwide (e.g. Kritikos et al., 2015; Nowicki Jessee et al., 2018; Tanyas et al., 2019), or a small sample of the affected area mapped immediately following the earthquake (e.g. Robinson et al., 2017). Here, we refer to these two model types as "global" and "same-event" models respectively. The global model of Nowicki Jessee et al. (2018) is routinely used to generate landslide predictions after large earthquakes, which are published on the United States geological survey (USGS) website https://earthquake.usgs.gov/data/ground-failure/. These products provide useful predictions of landslides triggered by earthquakes within hours of the event (e.g. Thompson et al., 2020). However, this model has been shown to struggle in the case of complicated events, for example in 2018, when landslides were triggered by a series of earthquakes in Lombok, Indonesia, rather than a single large event (Ferrario, 2019).

Several SAR methods have been developed for use in earthquake-triggered landslide detection based on the SAR amplitude (e.g. Ge et al., 2019; Konishi and Suga, 2018; Mondini et al., 2019) or interferometric SAR (InSAR) coherence (a pixel-wise estimate of InSAR signal quality) (e.g. Burrows et al., 2019, 2020; Olen and Bookhagen, 2018; Yun et al., 2015) or on some combination of the two (e.g. Jung and Yun, 2019). SAR data can be acquired in all weather conditions, and with recent increases in the number of satellites in operation, data are likely to be acquired within days of an earthquake anywhere on Earth. The removal of vegetation and movement of material caused by a landslide alters the scattering properties of the ground surface, giving it a signal in SAR data. Burrows et al. (2020) demonstrated that InSAR coherence methods can be widely applied in vegetated areas, and can produce usable landslide information within two weeks of an earthquake. However, in some cases false positives can arise from building damage or factors such as snow or wind damage to forests.

Recently, Masato et al. (2020) and Aimaiti et al. (2019) have demonstrated the possibility of combining SAR-based landslide indicators with topographic parameters in order to improve classification ability. Aimaiti et al. (2019) used a Decision Tree





method to combine topographic slope with SAR intensity and InSAR coherence to detect landslides triggered by the 2018 Hokkaido earthquake, and Masato et al. (2020) used Random Forest classification to combine several landslide indicators based on polarimetric SAR and topography to detect landslides triggered by two events in Japan: the 2018 Hokkaido earthquake and heavy rains in Kyushu, 2017. While these studies established the promise of a combined approach to landslide detection, they

did not assess the relative merits of empirical, SAR and combined methods. Furthermore, the two studies combined only SAR and topographic landslide indicators, omitting factors such as lithology, land cover and ground shaking data, which are also commonly used in empirical modelling of earthquake-triggered landslides (Nowicki Jessee et al., 2018; Robinson et al., 2017).

Here we aim to establish which of these three options is best: Landslide susceptibility maps, detection with InSAR coherence, or a combination of these. In order to do this, we began with an empirical model of landslide susceptibility based on ground

shaking, topography, lithology and land cover, all of which are available within hours of an earthquake. To this model, we then progressively added landslide indicators derived from InSAR coherence in the order that the SAR images became available following each case study earthquake. At each stage in this process, we assessed the ability of the model to recreate the landslide areal density (LAD) in the test area of the landslide dataset using receiver operating characteristic (ROC) analysis and by calculation of the coefficient of determination ($r^2$). For the modelling, we used Random Forests, a machine learning

technique that has been demonstrated to perform well in landslide detection (Chen et al., 2017; Fan et al., 2020). We chose to model LAD rather than individual landslide locations as both empirical models and SAR-based methods perform best at relatively coarse spatial resolutions (within the range 0.01-1 km$^2$ (Burrows et al., 2019; Nowicki Jessee et al., 2018; Robinson et al., 2017)). Similarly, the empirical models of landslide susceptibility released by the USGS following large earthquakes take the form of a predicted LAD, which can be interpreted as the probability for any location within the cell to be affected

by a landslide (Nowicki Jessee et al., 2018; Thompson et al., 2020). We used three case study earthquakes and assessed the effect of adding SAR-based landslide indicators to both the same-event model type and a preliminary global model, which was trained on two events and used to predict the third, allowing speculation on the performance of a global model trained on a larger number of earthquakes.

## 2 Data and methods

Empirical models of landslide hazard (represented here by LAD) adopt a functional form that is driven by a training dataset of mapped landslides rather than by the mechanics of slope stability. Since LAD is a continuous measure, our aim was to carry out a regression between a training dataset of mapped LAD and a selection of input features (e.g. slope, elevation, land cover) that influence landslide likelihood. The resultant function can then be used to predict LAD in areas where there are no mapped landslide data available based on these input features. In this section, we first describe the landslide data used to train and

test our models. Second, we describe the different input features which we used in the regression to predict LAD. Third we describe the Random Forests method and its implementation in this study. Finally we describe the metrics used to assess model performance.





## 2.1 Landslide data sets

We used polygon landslide inventories compiled for three large earthquakes that each triggered thousands of landslides: the
inventory of Roback et al. (2018) of 24,915 landslides triggered by the $M_w$ 7.8 2015 Gorkha, Nepal earthquake; the inventory
of Zhang et al. (2019) of 5,265 landslides triggered by the $M_w$ 6.6 Hokkaido, Japan earthquake; and the inventory of Ferrario
(2019) of 4,823 landslides triggered by the $M_w$ 6.8 Lombok, Indonesia earthquake on 5 August, 2018 (See Figure 1 for the
extent of triggered landslides from each of these earthquakes and the spatial and temporal coverage of the SAR data used
here). The performance of five SAR-based methods using the same SAR data used here has already been carried out for these
three case study earthquakes (Burrows et al., 2020). This allows a direct comparison between the performance of the models
developed in this study and that of existing SAR-based methods of landslide detection. Predicted LAD based on the empirical
model of Nowicki Jessee et al. (2018) was also available to download for these three events from the USGS website.

We converted the three polygon landslide inventories to rasters with a cell size of 20 × 22 m. We then calculated LAD within
10 × 10 squares of these 20 × 22 m cells, resulting in an aggregate landslide surface with a resolution of 200 × 220 m. This
is the same as the resolution at which Burrows et al. (2020) assessed SAR-based methods of landslide detection, allowing a
direct comparison with that study, and similar to the resolution of the model of Nowicki Jessee et al. (2018), whose products
are provided at a resolution of 0.002° (approximately 220 m, depending on latitude).

## 2.2 Training and test data sets

When developing an empirical model, it is necessary to divide the data into two parts: a training data set, which is used to
train the Random Forest; and a test data set, which is used to test model performance. Here we used two types of model setup:
"same-event" models trained on a small mapped area of an event to predict the landslide distribution across the rest of the
affected area (e.g. Robinson et al., 2017) and "global" models trained on historic landslide inventories to predict a new event
(e.g. Kritikos et al., 2015; Nowicki et al., 2014; Nowicki Jessee et al., 2018; Tanyas et al., 2019).

The real-world application of a same-event model is that a small number of landslides can be mapped manually from optical
satellite imagery in the days following the earthquake. The distribution of these landslides can then be used to predict the
landslide distribution across the whole affected area in much less time than would be required to manually map the whole area
(Robinson et al., 2017). For our same-event model, we randomly selected 250 landslide (LAD > 0.01) and 250 non-landslide
pixels for use as training data. Robinson et al. (2017) demonstrated that 250 landslide samples were sufficient to train their
landslide probability model. Since here we use 200 x 220 m pixels, 250 pixels is equivalent to a mapped area of around 11
km$^2$.

For the second approach, in which an empirical model is trained on a global inventory of past landslides, we trained the
Random Forest on two of our case study events and predicted the third. In this case, we randomly undersampled both the
landslide (LAD > 0.01) and non-landslide pixels to select 1000 landslide and 1000 non-landslide pixels from each event. The
resulting model was therefore trained on equal numbers of pixels from the two training events, rather than being dominated by
whichever was larger. We trained an ensemble of 100 models, performing the random undersampling independently each time



so that each model within the ensemble is trained on a different set of cells. We then estimated LAD using each model and took the median ensemble prediction as our final model. This process reduces variability and also allows calculation of upper and lower bounds for the LAD of every cell, as well as other statistical parameters such as the variance in predicted LAD.

It should be noted that successful global empirical landslide prediction models are trained on data from considerably more
case study events than this; for example, the model used by the USGS of Nowicki Jessee et al. (2018) was trained on 23 landslide inventories and the model of Tanyas et al. (2019) was trained on 25. Masato et al. (2020) demonstrated that when using fully polarimetric SAR data, it was possible to predict the spatial distribution of landslides triggered by the Hokkaido earthquake using a model trained on landslides triggered by a heavy rainfall event in Kyushu, 2017, but that it was not possible to predict landslides triggered by the Kyushu event using a model trained on Hokkaido. Therefore, given the small number of
case-studies used here to train our preliminary "global model", we expect a similar performance to that observed by Masato et al. (2020), i.e. we do not expect high performance on all predicted events. Instead, a high performance on at least one predicted event would be considered a success; this would suggest that such an approach is worthy of further investigation and is likely to improve when trained on a number of case-studies more comparable to those used by current "gold-standard" models (i.e. 23-25).

## 2.3  Input Features

A large number of possible input features have been used in previous work on landslide susceptibility mapping, including a wide range of topographic parameters, ground shaking estimates, rainfall data, lithology, landcover and distance to features such as rivers, roads and faults. Here we limited the model input features to globally available datasets in order to ensure the widest applicability of the results. We targeted our model at earthquake- rather than rainfall-triggered landslides, as these have
been more widely used as case studies when developing and testing SAR-based methods of landslide detection (e.g. Aimaiti et al., 2019; Burrows et al., 2019; Jung and Yun, 2019; Yun et al., 2015) and detailed polygon inventories of thousands of landslides triggered by a single earthquake (Ferrario, 2019; Roback et al., 2018; Zhang et al., 2019) provide a good source of test and training data for the model. In this section, we describe the topographic, ground shaking, land cover and lithology input features used to generate the initial same-event models. We also describe the model of Nowicki Jessee et al. (2018), which we
used as a base for the global models, and the SAR-based input features that we added to the same-event and global models. A summary of these inputs and which of our models they are used in is given in Table 1.

### 2.3.1  Topographic features

For input features derived from topography, we used the 30 m shuttle radar topography mission (SRTM) digital elevation model (DEM) (Farr et al., 2007). When processing the SAR data, this DEM was resampled to a resolution of 20 m × 22 m. From this
resampled DEM, we calculated slope and aspect using a 3 × 3 moving window and the compound topographic index (CTI), which provides a static proxy for soil moisture (Moore et al., 1991). We aggregated 10 × 10 grids of these cells to produce input features at a 200 m × 220 m scale. For each aggregate 200 × 220 m cell, we calculated: the mean 20 m × 22 m cell elevation (used by Catani et al., 2013; Nowicki et al., 2014), the standard deviation of cell elevations (used by Catani et al.,



2013), the maximum slope within an aggregate cell (used by Nowicki et al., 2014), the mean slope within an aggregate cell

(used by Nowicki Jessee et al., 2018; Robinson et al., 2017; Kritikos et al., 2015; Tanyas et al., 2019), the standard deviation of pixel slopes within an aggregate cell (used by Catani et al., 2013; Tanyas et al., 2019), the circular mean of the aspect (used by Chen et al., 2017), the mean compound topographic index (used by Nowicki Jessee et al., 2018; Tanyas et al., 2019) and the relief, or maximum elevation difference between all 20 m $\times$ 22 m pixels within the aggregate cell (used by Tanyas et al., 2019).

### 2.3.2 Ground shaking estimates


The inclusion of ground shaking information is what distinguishes the earthquake-specific prediction of triggered landsliding (used here) from a static estimate of landslide susceptibility (e.g. Nadim et al., 2006). Past studies have used Modified Mercalli Intensity (Kritikos et al., 2015; Tanyas et al., 2019), peak ground acceleration (Robinson et al., 2018; Nowicki et al., 2014; Tanyas et al., 2019) and peak ground velocity (PGV) (Nowicki Jessee et al., 2018; Tanyas et al., 2019). Here, we used PGV, as

this does not saturate at high shaking intensities (Nowicki Jessee et al., 2018).

An initial estimate of ground shaking is generally available within hours of an earthquake from the USGS ShakeMap webpage, and is then refined as more data become available. Allstadt et al. (2018) demonstrated that in some cases, such as the 2016 $M_w$ 7.8 Kaikōura, New Zealand earthquake, the version of the shaking estimate used can have a significant impact on the modelled landsliding. This was also observed in the case of the $M_w$ 7.1 2018 Anchorage, Alaska earthquake (Thompson et al.,

2020). Although our results are presented as a timeline where the model was revised as SAR data became available, we used the same version of PGV throughout, which is the final available model for each event. Our inital susceptibility model may therefore perform better than one that uses the data made available within the first few hours of the earthquake. However, the availability of SAR data for ground deformation measurements is one of the factors that can significantly improve the shaking estimate, so in practice, it will only be possible to add SAR-based landslide indicators to the final or close-to-final versions of

these empirical models. It is also preferable to isolate the effect of incorporating SAR by keeping all other input features the same in this study.

### 2.3.3 Lithology

Lithology is one factor that determines rock strength, and therefore landslide likelihood (Nadim et al., 2006), and has been used in several empirical landslide susceptibility models (Chen et al., 2017; Kirschbaum and Stanley, 2018; Nowicki Jessee

et al., 2018). We used the Global Lithological Map database (GLiM) (Hartmann and Moosdorf, 2012), which has 13 basic lithological classes, with additional "water bodies", "ice and glacier" and "no data" classes, which are supplied as polygon data. For each 200 x 220 m pixel, we took the dominant basic lithological class (i.e. that with the largest area share), resulting in a categorical input feature. One advantage of Random Forests is their ability to use both continuous and categorical input features, but empty categories in the training data can lead to biases in the model (Au, 2018). To avoid this, we used the

"onehot" method, in which each category is supplied to the model as a separate dummy input feature i.e. a binary surface of, for example, "unconsolidated sediment" and "everything else" (Au, 2018). The number of input feature maps used for each





model was equivalent to the number of lithological categories present in each case: six in Hokkaido, nine in Nepal and six in Lombok.

### 2.3.4 Land cover

Nowicki Jessee et al. (2018) used land cover as a proxy for vegetation coverage and type, as the composition of the soil and the presence or absence of plant roots can affect slope stability. Here we used land cover data downloaded from the European Space Agency (ESA) Climate Change Initiative, which includes yearly maps of 22 land cover categories compiled at a 300 m resolution from 1992-2015 (ESA, 2017). We used the 2014 map as the most recent land cover map preceding all of our case study events. Like lithology, land cover is a categorical variable, thus we used the same "onehot" method described in Section

2.3.3 to avoid biasing due to empty categories, and selected the dominant land cover type within each 200 m x 220 m aggregate cell. As for lithology, the number of land cover input feature maps used for each case study area was equivalent to the number of categories present: 15 in Hokkaido, 16 in Nepal and 16 in Lombok.

### 2.3.5 USGS ground failure product

For our global model, in which two events were used as training data and the third was predicted, we found that it was not

possible to generate a reasonable initial model using the landslide influencing factors listed in Sections 2.3.1-2.3.4. The models generated in this way had limited skill in Hokkaido and no skill in Lombok or Nepal, which is not representative of the performance of existing global models (e.g. Nowicki Jessee et al., 2018; Tanyas et al., 2019). This poor performance is likely to be due to the limited number of case studies used as training data in our global models and would make it difficult to draw conclusions on the benefits of adding SAR to a global empirical model. Therefore, instead of using the set of individual input

feature maps in the global model, we used one single feature map that encapsulates the likely best results from a global model trained across a large number of events: the output USGS Ground Failure product of Nowicki Jessee et al. (2018). We found that by replacing the individual input features with the model output of Nowicki Jessee et al. (2018), we improved model performance in Nepal and Lombok, and obtained a more consistent result across the three events.

The models of Nowicki Jessee et al. (2018) are published within hours following a large earthquake for use in hazard

assessment and emergency response coordination and are available from the USGS website. They are generated using Logistic Regression based on PGV, Slope, CTI, Lithology and Land Cover data (Nowicki Jessee et al., 2018). As described in Section 2.3.2, these models evolve in the days and weeks following an earthquake as the best available PGV estimate is updated. Here we used the final version of each model, which may perform better than the model version that would be available within hours of an earthquake. However, any conclusions drawn here about the advantage of incorporating SAR into the model should

remain valid for earlier versions of the ground failure product. The models of Nowicki Jessee et al. (2018) are published at a similar spatial resolution (around 0.002°) to our models. Therefore the only processing step required was to resample them onto the geometry of our other input data (200 x 220 m pixels).



### 2.3.6 InSAR coherence features (ICFs)

Multiple studies have demonstrated that methods based on InSAR coherence can be used in landslide detection (e.g. Aimaiti
et al., 2019; Burrows et al., 2019, 2020; Jung and Yun, 2019; Yun et al., 2015). An interferogram is formed from two SAR
images acquired over the same area at different times, and its coherence is sensitive to a number of factors including changes to
scatterers between the two image acquisitions and, particularly in areas of steep topography, changes to the image acquisition
geometry (Zebker and Villasenor, 1992). Changes in soil moisture (Scott et al., 2017), movement of vegetation due to wind
(Tanase et al., 2010), growth, damage or removal of vegetation (Fransson et al., 2010) and damage to buildings (Fielding et al.,
2005; Yun et al., 2015) are all examples of processes which can alter the scattering properties of the Earth's surface and so lower
the coherence of an interferogram. InSAR coherence is a measure of the signal-to-noise ratio of an interferogram estimated on
a pixel-by-pixel basis from the similarity in amplitude and phase change of small numbers of pixels (Just and Bamler, 1994).

We used SAR data from two SAR systems in this study: the C-band Sentinel-1 SAR satellite operated by ESA and the
Phased Array type L-band SAR-2 (PALSAR-2) sensor on the Advanced Land Observation Satellite 2 (ALOS-2) operated
by the Japanese Space Agency (JAXA). For each case study earthquake, we used ascending and descending track Sentinel-1
SAR imagery and a single track of ALOS-2 PALSAR-2 imagery (Figure 1). This volume of SAR data is available in the
immediate aftermath of the majority of earthquakes (Burrows et al., 2020). These two satellite systems acquire SAR data at
different wavelengths, and so interact slightly differently with the ground surface, with ALOS-2 generally being less noisy in
heavily vegetated areas than Sentinel-1 (Zebker and Villasenor, 1992). SAR data are acquired obliquely, which in areas of steep
topography can lead to some slopes being poorly imaged by the SAR sensor. Here, descending-track data were acquired with
the satellite moving south and looking west, while ascending track were acquired with the satellite moving north and looking
east. Therefore, slopes which are poorly imaged in ascending data are likely to be better imaged by the descending track and
vice versa. It is for this reason that we employed two tracks of Sentinel-1 data, but this is currently not often available for L-
band SAR. We processed the SAR data using GAMMA software, with Sentinel-1 data processed using the LiCSAR package
(Lazeckỳ et al., 2020). The data were processed in a Range × Azimuth coordinate system and then projected into a geographic
coordinate system with a spatial resolution of 20 m × 22 m. Further details on parameter choices made in the generation of the
CECL, Bx-S, PECI, $\Delta C\_sum$ and $\Delta C\_max$ surfaces can be found in Burrows et al. (2020).

We aimed to combine InSAR coherence methods with empirical models. To achieve this, we used multiple InSAR coherence
methods as input features. The first coherence method we used was the coherence of the co-event interferogram (formed from
two images spanning the earthquake). The movement of material and removal of vegetation by a landslide alters the scattering
properties of the Earth's surface, resulting in low coherence. This gives landslides a low coherence in a co-event interferogram.
Burrows et al. (2019) and Vajedian et al. (2018) demonstrated that this had some potential in triggered landslide detection.
Second, we used the coherence of the post-event interferogram (formed from two images acquired after the earthquake).
Coherence is dependent on the land cover, with vegetated areas generally having a lower coherence than bare rock or soil
(Tanase et al., 2010). Therefore the bare rock or soil exposed following a landslide is likely to have a higher coherence than the
surrounding vegetation. This input feature was found to be beneficial in most cases, but the distribution of post-event coherence





values varied significantly between events when using ALOS-2 data. This may be because the wait time between the first and second post-event images varied significantly between the events (from 14 to 154 days, Burrows et al., 2020). The post-event ALOS-2 coherence was therefore omitted from our global models.

As well as the raw coherence surfaces, we used the five coherence-based methods tested by Burrows et al. (2020) as input features, all of which showed some level of landslide predictive skill in that study. First, the co-event coherence loss (CECL) method uses the decrease in coherence between a pre-event and co-event interferogram to identify "damaged" pixels (Fielding et al., 2005; Yun et al., 2015). The Boxcar-Sibling (Bx-S) method of Burrows et al. (2019) uses the difference between two different co-event coherence estimates to remove large spatial scale coherence variations from the landslide detection surface.

Finally three methods presented by Burrows et al. (2020) incorporate the coherence of a post-event interferogram to detect landslides, making use of the fact that the coherence decrease caused by a landslide is temporary. The post-event coherence increase (PECI) method uses the difference between the post-event and co-event coherences. The PECI and CECL methods are then combined in two further methods: the sum of the co-event coherence loss and post-event coherence increase ($\Delta C\_sum$); and the maximum coherence change ($\Delta C\_max$), where for every pixel, whichever is largest of PECI or CECL is taken.

The majority of the methods outlined in this section use a "boxcar" coherence estimate, using pixels from within a 3 x 3 window surrounding the target pixel in the coherence estimation. The only exception to this is the Bx-S method, which also uses a "sibling" coherence estimate in which an ensemble of pixels is selected from within a wider window for coherence estimation using the RapidSAR algorithm of Spaans and Hooper (2016) (Burrows et al., 2019). This sibling coherence estimation requires additional data (a minimum of six pre-seismic images), and so it was not possible to calculate this using ALOS-2 data for the

2015 Nepal earthquake, as this event occurred very early in the lifetime of the satellite. The Bx-S method with ALOS-2 data was therefore not used in Nepal, or in any of the global models we tested.

## 2.4   Random Forest theory and implementation

Random Forests are an extension of the Decision Tree method, a supervised machine learning technique in which a sequence of questions are applied to a data set in order to predict some unknown property, for example to predict landslide susceptibility

based on features of the landscape (e.g. Chen et al., 2017). A Random Forest comprises a large number of decision trees, each seeing different combinations of input data, which then make a combined prediction. This avoids overfitting the training data, a problem when using individual decision trees (Breiman, 2001). A very simple example of the Random Forest method using only two trees to estimate the value a sample should be assigned based on two input features (the colour and shape of each sample) is shown in Figure 2. First the training dataset is bootstrapped so that each tree sees only a subset of the original pixels

(Figure 2a). Each tree carries out a series of "splits", in which the data are divided in two based on the value of an input feature (e.g. sample shape or colour in Figure 2). These splits are chosen based on the improvement they offer to the ability of each tree to correctly predict its training data. Every tree remembers how it split the training data, and then applies the same splits as it attempts to model the test data (Figure 2c). For Random Forest Regression, the mean value of all trees is taken as the model output for every sample (Breiman, 2001).





Here, we used Random Forests to carry out the regression between the input features described in Section 2.3 and LAD. Random Forests are well suited to the combination of SAR methods with static landslide predictors for several reasons. Random Forests are relatively computationally inexpensive, and because of this, can use a large number of input features. Random Forests do not require input features to be independent, which is advantageous here since InSAR coherence is sensitive to both slope and land cover, as well as the presence or absence of landslides. Input features of Random Forests do not need to be

monotonic and can be categoric or continuous. Catani et al. (2013); Chen et al. (2017); Fan et al. (2020) and Masato et al. (2020) have demonstrated that Random Forests can yield good results in landslide prediction.

To implement the Random Forests method, we used the Python scikit-learn package (Pedregosa et al., 2011). The model is defined by a number of hyperparameters that can have a noticeable effect on the model. First of these is the criteria on which a split should be assessed. In our models, splits were carried out that minimised the mean absolute error (MAE). This

criterion was selected based on Ziegler and König (2014). The second hyperparameter describes the bootstrapping step (e.g. Figure 2a). Here the data were bootstrapped so that a number of random samples was taken equal to the number of pixels in the training dataset. Each individual pixel is therefore likely to appear at least once in around two thirds of the bootstrapped datasets (Efron and Tibshirani, 1997). This process improves the stability of the model by ensuring that each tree is trained on a slightly different subset of the training data (Breiman, 2001).

Table 2 shows a further five hyperparameters that define the setup of a Random Forest model in scikit-learn. First the number of trees, n_estimators, defines the number of decision trees that make up a forest. More trees can increase model accuracy up to a point, but also result in a more computationally expensive model. Second, max_features defines the fraction of possible input features considered when selecting how to split the data is calculated. For example, our initial same-event models had 11 input features (Table 1), so with square root (sqrt) selected, the model will assess possible splits based on $\sqrt{11} \approx 3$ input

features before identifying the best split. Max_depth defines the maximum "depth" of each decision tree i.e. the longest path length from the beginning of the tree to the end. For example, in Figure 2, Tree 1 has a depth of 3 and Tree 2 has a depth of 2. Finally, min_samples_split is the minimum number of samples a node has to contain before splitting for a split to be allowed, while min_samples_leaf is the minimum number of samples assigned to either branch after splitting for a split to be allowed. For each of these five hyperparameters, we selected several possible options, which are shown in the final column

of Table 2. We then used the GridSearchCV function in sci-kit learn to select an optimised model. This function ran models with every possible combination of these values, using 4-fold cross-validation over our training data to identify the optimal hyperparameter combination (Pedregosa et al., 2011).

### 2.4.1 Feature importances

The importance of each input feature was calculated from the decrease in MAE resulting from splits on that feature for each

tree, then averaged over all the trees to obtain feature importance at forest level. This calculation gives an indication of how reliant the model is on each feature, with the sum of importances across all input features equal to one (Liaw et al., 2002). Feature importance therefore helps with interpreting the model and can allow unimportant features to be eliminated from future models, reducing computation time (e.g. Catani et al., 2013; Díaz-Uriarte and De Andres, 2006). However, it should be





noted that the importance of categorical input features (here lithology and landcover), is often underestimated, since in this

case the number of possible splits is limited to the number of categories present in the training data. Furthermore, since our input features are not independent (for example, InSAR coherence can be affected by land cover and slope), caution should be taken when drawing conclusions from importance values.

## 2.5   Performance metrics

Each model generated a raster of continuous predictor variables, corresponding to the modelled LAD in the range [0,1]. We

assessed model performance by comparing the test areas of these predicted surfaces with the mapped LAD calculated in Section 2.1. We used two metrics in assessing model performance: ROC analysis and the coefficient of determination ($r^2$).

ROC analysis has been widely used in studies of landslide prediction and detection and is relatively simple to interpret (e.g. Burrows et al., 2020; Robinson et al., 2017; Tanyas et al., 2019). Additionally, the use of ROC analysis allows comparison between the models in this study and the InSAR-coherence-based methods of Burrows et al. (2020). ROC analysis requires

a binary landslide surface for validation, so we applied a threshold to the mapped LAD surface calculated in Section 2.1, assigning aggregate cells with LAD > 0.1 as "landslide" and < 0.1 as "non-landslide". Setting this threshold higher would test the models ability to detect more severely affected pixels, while setting it lower would test the models ability to more completely capture the extent of the landsliding. We chose 0.1 to strike a balance between these two factors. These aggregate cells with LAD > 0.1 contain 77% of the total area of the individual landslide polygons in Nepal, 94% in Hokkaido and 46%

in Lombok. Using this binary aggregate landslide surface, the false positive rate (the fraction of mapped non-landslide pixels wrongly assigned as "landslide" by the model) and true positive rate (the fraction of mapped landslide pixels correctly assigned as "landslide" by the model) were calculated at a range of thresholds and plotted against each other to produce a curve. The area under this curve (AUC) then indicates the predictive skill of the model, with a value of 0.5 indicating no skill and 1.0 indicating a perfect model (Hanley and McNeil, 1982). The ROC AUC values calculated here therefore represent the ability of

the model to identify pixels with LAD > 0.1.

The second method we used to assess model performance was to carry out a linear regression between the observed and predicted LAD. The $r^2$ of this regression is calculated as the fraction of the variability in predicted LAD that is explained by this linear model. A high $r^2$ coefficient value (up to a maximum of 1.0) indicates low levels of random errors and suggests that the model fits the observed LAD well and an $r^2$ value of zero represents a model with no skill. Therefore, $r^2$ indicates the

ability of our Random Forest models to avoid random errors in their prediction of LAD, while the ROC AUC indicates their ability to identify severely affected pixels.





## 3   Results

### 3.1   Same-event models

Figure 3 shows the effect on model AUC (panels a, b, c) and $r^2$ (panels d, e, f) of adding Sentinel-1 and ALOS-2 data to a
landslide susceptibility model trained on 250 landslide and non-landslide pixels mapped following an event, and tested on the
remaining pixels. Since the model performance varied depending on which 500 pixels were used as training data, we ran each
model 30 times and give the mean AUC and $r^2$ of these models in Figure 3(a-f).

The initial models for Hokkaido, Nepal and Lombok had AUC values of 0.60, 0.75 and 0.79 and $r^2$ values of 0.17, 0.027 and
0.020 respectively. We observed this combination of comparatively high ROC AUC values and very low $r^2$ values for all of the
models tested in this paper as well as for the models of Nowicki Jessee et al. (2018). These low $r^2$ values have implications for
how the current generation of empirical models should be interpreted and we discuss this further in Section 4.1.

In almost all cases, incorporating ICFs improved model performance in terms of AUC, with the biggest improvement ob-
served when ICFs from the first ALOS-2 image are added. The addition of these first ALOS-2 ICFs also results in an increase
in $r^2$ for each case study region. The biggest improvement is seen in Hokkaido, where the addition of these ALOS-2 ICFs
results in a increase in $r^2$ from 0.17 to 0.45 and an increase in AUC from 0.68 to 0.78. Although these ALOS-2 ICFs result
in the largest improvement in model performance, in all three events the model incorporating both ALOS-2 and Sentinel-1
ICFs outperforms the model using ALOS-2 ICFs alone in terms of AUC. In terms of $r^2$, the combined model outperforms
the ALOS-2 only model in Nepal, but the ALOS-2 only model performs best in Lombok. In Hokkaido, the combined model
performs best after the third Sentinel-1 acquision. Sentinel-1 data are also often available sooner after a triggering earthquake
than ALOS-2 data, due to the short revisit time of the Sentinel-1 satellites.

### 3.2   Global models

An alternative to training on a small part of the affected area is to train the model on inventories from past earthquakes. To
test the effect that incorporating ICFs might have on such models using our three case studies, we trained models on two
earthquakes and predicted the LAD triggered by the third. Figure 3 shows the effect on model AUC (panels g, h, i) and $r^2$
(panels j, k, l) of adding ALOS-2 and Sentinel-1 ICFs to the input features used in training the model.

The USGS landslide model of Nowicki Jessee et al. (2018) had AUC values of 0.65 in Hokkaido, 0.62 in Lombok and 0.77
in Nepal and $r^2$ of 0.026, 0.033 and 0.0022 respectively. In order to test the effect of adding ICFs to a global model, we used
the USGS model as an input feature alongside the ICFs listed in Section 2.3. As with the same-event models (Section 3.1),
incorporating ICFs improved model performance in almost all cases with the largest improvement in terms of both AUC and
$r^2$ seen with the addition of the first ALOS-2 ICFs. In all three cases, our global models at 2 weeks have an AUC > 0.8 and
outperform the USGS landslide model of Nowicki Jessee et al. (2018) in terms of AUC and $r^2$. Overall, the addition of ICFs
derived from SAR data acquired within 2 weeks of each earthquake results in better and more consistent model performance.
This improvement can be observed in Figure 5, which shows the models of Nowicki Jessee et al. (2018) alongside those
incorporating SAR data acquired within 2 weeks of each earthquake.





### 3.3 Do these models outperform individual InSAR coherence methods?

To assess whether a combined model of ICFs and landslide susceptibility is useful, it is necessary to compare it to the information that could be obtained from SAR alone. The SAR data and case studies used here are the same as those used by Burrows et al. (2020) in their systematic assessment of InSAR coherence methods for landslide detection. AUC values have been calculated here at the same resolution and with the same definition of a landslide or non-landslide aggregate cell. This allows direct comparison between the performance of the models applied here and the InSAR coherence methods alone. To assess the value added to the InSAR coherence methods by the static landslide predictors (e.g. slope), we also applied the Random Forests technique using the ICFs described in Section 2.3.6 as the only input features. Figure 6 shows AUC values for 1) the combined model, 2) a SAR only model and 3) the InSAR coherence-based method recommended for each SAR image acquisition by Burrows et al. (2020, Supplement).

For the same-event models (Figure 6, panels a-c), the combined model outperforms the SAR-only model in almost all cases and in the initial days following the earthquake, when only Sentinel-1 is available, the combined model outperforms the coherence methods. This is particularly noticeable for Lombok, as 4 Sentinel-1 images were acquired before the first ALOS-2 image following this event, and the model consistently has an AUC of around 0.2 more than the individual Sentinel-1 coherence surfaces. When the coherence methods recommended by Burrows et al. (2020) are employed using ALOS-2 data, they outperform both the combined and SAR-only models in terms of AUC in two cases (the first ALOS-2 image after Hokkaido and the second after Nepal) and perform similarly in the other four cases.

For the global models, the difference between the SAR-only and combined model is less pronounced, but in most cases, the combined model has a higher AUC than the SAR-only model. Both models outperform the individual InSAR coherence methods that use Sentinel-1 data in all cases except the first Sentinel-1 image following the Nepal Earthquake. The coherence methods of Burrows et al. (2020) outperform both models when the second ALOS-2 image is available in Nepal and Lombok, but have a similar performance in the other cases.

### 3.4 Feature importances

Figure 7 shows the relative importance of each input feature in a Random Forest model when the model is trained on data from Hokkaido (a), Nepal (b), and Lombok (c). For simplicity, the parameters have been grouped by the data required to create them. For example, the importance of "topography" on Figure 7 was calculated as the sum of the importances of maximum slope, mean slope, standard deviation of slope, elevation, standard deviation of elevation, relief, CTI and aspect. For each case study, topography was initially the most important input feature but its importance gradually decreased as more SAR data were added, particularly in Hokkaido, where it ceased to be the most important feature after the ICFs from the first ALOS-2 image were added to the model. In the final models, ALOS-2 ICFs were consistently more important than Sentinel-1, particularly in Hokkaido where these become the most important feature in the model.





## 4    Discussion

We have presented the results of adding ICFs to two types of landslide model: a same-event model trained on a small area of each case study earthquake, and a global model, which is trained on two earthquakes to predict the third. In both cases, we have demonstrated that model performance was significantly improved by the addition of ICFs derived from data acquired within 2

weeks of an earthquake. In this section, we discuss some of the factors that could affect the applicability of these models in an emergency response situation.

### 4.1    Model interpretation based on ROC AUC and $r^2$

We have demonstrated that the addition of ICFs to empirical models of LAD based on topography, ground shaking estimates, land cover and lithology significantly improves their performance in terms of ROC AUC and $r^2$. However, while the final AUC

values are around 0.8 or higher, $r^2$ remains below 0.05 in Nepal and Lombok. We also observed this combination of relatively high AUC (0.63-0.77) alongside low $r^2$ (0.0002-0.03) for the models of Nowicki Jessee et al. (2018), which are published on the USGS website. This result directly impacts how the current generation of empirical models at this spatial scale should be interpreted. The low $r^2$ values we have observed indicate that the ability of the models to predict LAD as a continuous variable is poor. The more encouraging AUC values indicate that the models are well-suited to discriminating between affected and

unaffected pixels. Therefore, in an emergency response scenario, the model of Nowicki Jessee et al. (2018) and the models presented here can be used to provide an estimate of the spatial extent and distribution of the triggered landsliding after an earthquake, but should not be interpreted as a reliable estimate of LAD.

### 4.2    Selection of the training data for the same-event model type

The same-event case we have presented here uses 500 training data cells randomly selected from across the study area. The

aim of this process is to reduce the area that is required to be manually mapped from optical satellite data or field investigations before an estimate of LAD can be generated over the whole affected area, and therefore to reduce the time taken to generate a complete overview of the landsliding. An alternative scenario would be the case where cloud cover prevents manual landslide mapping in some parts of the affected area. Robinson et al. (2017) noted that this clustering had a detrimental effect on the performance of their same-event model.

To explore this, we tested the ability of a same-event model to predict LAD for a selected area using only training data from outside that area (Figure 8). We chose to test this in Nepal, as this event covered the largest area. We began with a training data set comprising all of the data outside the test area (Figure 8d), which contained 5210 cells with LAD > 1% across an area of 47 km$^2$, so that after balancing by undersampling non-landslide cells (see 2.4), the training data set comprised 10,420 cells. The model shown in (e) used only the data west of the training area (6,106 pixels after balancing across an area 21 km$^2$). From

here, the longitudinal extent of the training data used was halved each time, resulting in balanced training data sets of 3,728 (f), 1466 (g), 538 (h) and 180 (i) cells.





Figure 8 clearly shows that models trained on a smaller area have lower values of AUC (panel a) and $r^2$ (panel b), which is to be expected. $r^2$ increased in most cases as ICFs were added, with the exception of model (i), which was trained on only 180 cells. All models improved in terms of both AUC and $r^2$ when the first ALOS-2 ICFs were added. As moreICFs were

added to the model, AUC improved for all models, and the difference in AUC between the best and worst models decreased. An alternative way of viewing this is that the inclusion of ICFs in such models decreases the area required to be mapped before an obscured area can be modelled with the same AUC. For example, the model trained on all the data outside of the test area (10,420 mapped cells) achieved an AUC > 0.7 before the ICFs were added. Using ICFs derived from two Sentinel-1 images and one ALOS-2 image, model (h) also achieves an AUC > 0.7 with only 538 mapped cells. Panels (d-i) show the model using

ICFs derived from all SAR data acquired in the first 2 weeks following the earthquake (two Sentinel-1 images and one ALOS-2 image). In the locations where landslides were observed, the model signal strength visibly increases from (i-d) as the training area is increased, and in most cases, noise in the northern area of the model decreases.

The initial model in Figure 3b,e performs significantly better than those in Figure 8 according to ROC analysis, with an AUC of 0.75 compared to AUCs ranging from 0.54-0.74 from models (d-i) in Figure 8, despite the fact that models (d-g) are trained

on considerably more data (1566-10420 cells compared to 500). However, the improvement in the model when the ICFs were added is greater in terms of AUC in Figure 8a than in Figure 3b. For the model trained on 500 randomly selected pixels, mean AUC increases from 0.75 to 0.78 when ICFs from SAR data acquired within 2 weeks of the earthquake are added to the model. The AUC of model (h) in Figure 8, which is trained on a similar number of pixels (538) improves from 0.62 to 0.77 when the same ICFs were added. Therefore, it appears that the benefits to same-event models offered by the addition of ICFs may be

particularly relevant when cloud cover obscures part of the affected area.

## 4.3   Training data format

Here, we trained our same-event model on LAD calculated from polygon landslide data. In some cases, polygon landslide data may be available following an earthquake, for example the Geospatial Information Authority (GSI) Japan released a preliminary landslide map one week following the 2018 Hokkaido earthquake (GSI Japan, 2018). However, in the majority

of cases, landslides are more likely to be mapped as points rather than polygons in order to produce maps more quickly (e.g. Kargel et al., 2016; Williams et al., 2018). Therefore, the polygon landslide data we use here may not always be available soon enough following an earthquake to be used in training empirical models of landsliding.

The simple regression we have carried out here to recreate LAD would not be possible using point-mapped landslide inventories. The first alternative to this would be to carry out regression on landslide number density (LND) instead. LND and

LAD are generally well correlated following an earthquake (Cui et al., 2020; Ferrario, 2019), therefore we expect that the improvement observed here when ICFs are added to the model should also be observed when the regression is carried out for LND. The second alternative, which is employed by Nowicki Jessee et al. (2018) and Robinson et al. (2017), is to train the model at the mapping resolution, use machine learning to estimate the probability of landslide occurrence and then to carry out an additional regression step to convert this to an estimate of LAD at a lower spatial resolution. For example, Nowicki Jessee

et al. (2018) use the "two-buffer" method of Zhu et al. (2017), in which point-mapped landslides are used as the "landslide"





part of the training dataset. Non-landslide training data points are selected that are far enough from any landslide point to be unlikely to lie within a landslide, while being close enough that they are likely to have been included in the area mapped when the inventory was generated. This technique could be employed for models incorporating ICFs in the case where only point-mapped landslide data are available after an earthquake.

**4.4   Towards a global landslide prediction model**

The advantage of a global landslide prediction model, such as those of Nowicki Jessee et al. (2018) and Tanyas et al. (2019), is that there is no need to map any of the landslides triggered by an earthquake before a model of their likely spatial distribution and impacts can be produced. Our global models incorporating Sentinel-1 and ALOS-2 ICFs outperform our same-event models in Nepal and Hokkaido in terms of AUC. All of the model inputs used here are available globally, and thus could be

extended to a global model, but our "global" models were trained on only two events. It is likely that the model performance would improve if the model was trained on more events. In comparing their model, which was trained on 23 events, to a preliminary version trained on only 4 events (Nowicki et al., 2014), Nowicki Jessee et al. (2018) observed a noticeable improvement in performance for the model trained on more events.

Random Forests are not capable of extrapolation, meaning that if the predicted event has LAD values or input feature values

outside the range of those present in the training dataset, these will not be well modelled. A model trained on more events will cover a wider range of input feature and LAD values. Furthermore, although the model of Nowicki Jessee et al. (2018), which we used as an input feature to represent the combined effects of lithology, land cover, slope, CTI and PGV on landslide likelihood, was trained on a global set of landslide inventories, the combined effect of this model and the ICFs was modelled on only two events. It is therefore likely that the relationship between the original factors, the ICFs and landslide probability

could be better captured by a model trained on more landslide inventories, particularly since the model of Nowicki Jessee et al. (2018) did not perform well in Hokkaido or Lombok. This could have had an adverse effect on our models of Nepal, which were trained on these two events. This may explain the relatively poor performance of the combined Sentinel-1 models in Nepal compared to the performance of the model of Nowicki Jessee et al. (2018) in this location.

Despite only using two events as training data, the models incorporating ICFs from at least one ALOS-2 image outperformed

the original model of Nowicki Jessee et al. (2018) in all three events, resulting in an improvement in AUC of over 0.2 in Hokkaido and Lombok and in AUC > 0.8 in all three cases. This suggests that the addition of ICFs would improve both the performance and robustness of global models and would be particularly useful in the case of complicated events, such as Hokkaido, where landslides were triggered by both rainfall and an earthquake (Zhang et al., 2019). Model performance may improve further when trained on a larger number of case study events. It may also be that when a greater number of case study

events is used as training data, the addition of Sentinel-1 ICFs to the model will offer a greater improvement. However, for models trained on only two events, the addition of Sentinel-1 ICFs did not consistently improve the model.



### 4.5 Current recommendations for best practice

Here we have tested an established global model from Nowicki Jessee et al. (2018), InSAR coherence methods from Burrows et al. (2020) and a set of same-event and preliminary global models incorporating ICFs for three large earthquakes. We are
therefore able to make some recommendations on which of these options is most applicable at different stages following an earthquake. In Section 4.1, we argued that empirical models are best suited to distinguishing between affected and unaffected cells. For this application, the AUC values in Figures 3 and 8 are most relevant and our recommendations are based on this metric.

The model of Nowicki Jessee et al. (2018) initially has the highest AUC. After the first ALOS-2 image is acquired, the
performance of the same-event and global models is similar in Lombok and Nepal, but the global model performed considerably better than the same-event model in Hokkaido. All models using these first ALOS-2 ICFs outperformed the model of Nowicki Jessee et al. (2018). Therefore at this stage we would recommend adding the ICFs derived from ALOS-2 data or from a similar L-band SAR satellite to a global model.

In all of the cases here, Sentinel-1 data was acquired earlier after the earthquake than ALOS-2 data, resulting in a brief period
where only ICFs from Sentinel-1 could be added to the model. During this period, our same-event models outperform our global models in all three cases. The same-event models incorporating Sentinel-1 ICFs also outperform the model of Nowicki Jessee et al. (2018) by 0.18 in Lombok and 0.03 in Hokkaido (The two models have the same AUC in Nepal). Therefore there might be an advantage to applying a same-event model using Sentinel-1 ICFs before the first L-band SAR image is acquired. However, this would only be possible when sufficient landslide data can be generated in this time period to train the same-event
model, something which is likely to rely on cloud-free weather conditions and the availability of personnel to manually map the landslides. This is therefore only likely to be relevant in cases where the first L-band image is acquired relatively late, as it was following the Lombok earthquake studied here. In most cases, we do not expect this to happen, particularly as the number of L-band SAR satellites is set to increase in the near future (Section 4.6).

In some cases, the individual InSAR coherence methods of Burrows et al. (2020) had a higher ROC AUC than our models
or those of Nowicki Jessee et al. (2018) (Figure 6). However in practice, switching from an empirical model of landslide likelihood in the first few days of an emergency response to a different product derived from coherence would cause unnecessary confusion. It is preferable to have a single product that can be updated as more information becomes available. For this reason, our overall recommendation is that the model of Nowicki Jessee et al. (2018) should be used until the first L-band SAR image is acquired, and then ICFs derived from this image should be incorporated into a combined model that exploits the advantages
of both SAR and empirical landslide-mapping approaches.

### 4.6 Future possible SAR inputs

Although here we limited the SAR inputs to coherence-based landslide detection methods, there are other SAR-derived inputs that could be beneficial. First, the amplitude of the SAR signal has been shown to both be sensitive to landslides and skillful in their detection (Ge et al., 2019; Konishi and Suga, 2018, 2019; Mondini et al., 2019). As amplitude is not calculated from




multiple pixels, it can be used to obtain information at a higher spatial resolution than coherence. There have been three studies that successfully combined surfaces derived from amplitude and coherence to detect landslides triggered by the Hokkaido earthquake (Aimaiti et al., 2019; Ge et al., 2019; Jung and Yun, 2019), suggesting that amplitude methods may be beneficial to our model. However, amplitude depends on several factors including soil moisture content and slope orientation relative to the satellite sensor. Landslides can therefore result in both increases and decreases in amplitude, and false positives can easily
arise. Amplitude change is thus not usually reliable in landslide detection (Czuchlewski et al., 2003; Park and Lee, 2019).

A second option would be polarimetric SAR, in which data are recorded by the SAR satellite at two polarisations (dual-pol) or both projected and recorded at two polarisations (quad-pol). Polarimetric SAR data thus describe the scattering properties of the Earth's surface more completely than the single-polarisation data we have used here. It has been demonstrated that quad-pol data can be used to map landslides (Czuchlewski et al., 2003; Masato et al., 2020; Park and Lee, 2019). However, few SAR
systems acquire quad-pol data, and those that do (e.g. ALOS-2) do not acquire them routinely. Therefore these data are often not available immediately after an earthquake and so are not suitable for use in a global model. The applicability of dual-pol SAR data to landslide-detection is less well explored, but as these data are acquired more commonly and are routinely acquired by Sentinel-1 they are likely to be available following an earthquake and so could be incorporated. When testing a range of polarimetric parameters that could be used in landslide detection, Park and Lee (2019) demonstrated that landslides could be
identified in surfaces generated from dual-pol data, although using dual-pol methods resulted in more noise than quad-pol. Thus, when combined with the other landslide detection and susceptibility parameters used in this study, dual-pol Sentinel-1 data may offer an additional benefit.

It is also worth considering which SAR satellites could provide data to be incorporated into empirical models. Here we used two tracks of Sentinel-1 data and one of ALOS-2. ALOS-2 data are acquired at a longer wavelength than Sentinel-1 (L-band
rather than C-band), so the data retain a higher coherence in vegetated areas, a significant advantage in landslide mapping. It is therefore unsurprising that the addition of the first ALOS-2 ICFs to the models offered the greatest improvement to prediction of LAD. We have recommended that when the first ALOS-2 image becomes available after an earthquake, the ICFs derived from this image should be incorporated into a global model. The ALOS-2 satellite system has a 14-day repeat time and one of the aims of the mission is to enable emergency response, so the first ALOS-2 image is very likely to be available within
two weeks of an earthquake. In the future, we expect that it will also be possible to derive useful ICFs from L-band SAR data acquired by the planned NASA-ISRO NiSAR mission, which is due to launch in 2022. The NiSAR mission will acquire data continually with a 12 day repeat time (Sharma, 2019). Therefore, after the launch of this satellite system, it should be possible to use both ascending and descending tracks of L-band SAR data, rather than just one track as we have used here. NiSAR will also acquire SAR data at a wider swath width (240 km) than the ALOS-2 data we have used here (50 - 70 km) (Sharma, 2019),
which is an advantage for earthquakes that trigger landslides across a large area. For example, in the Nepal earthquake, the ALOS-2 scene we used in this study covers less than half of the landslides that were triggered by the earthquake (Figure 1a). Therefore, in order to obtain a complete model of a landslide event of this spatial extent, it would be necessary to combine several adjacent scenes, which are likely to be acquired on different days. ALOS-2 PALSAR-2 data are also acquired at a wider swath width (up to 490 km), but this is at the expense of spatial resolution, which coarsens to 100 m per pixel. Since coherence



is calculated over a $3 \times 3$ window, the resolution at which landslides could be detected would therefore decrease to 300 m. Therefore we expect that data acquired by the NiSAR mission could be extremely useful in generating ICFs.

### 4.7 Possible application to rainfall-triggered landslides

Here we began with a set of models that predicted the spatial distribution of earthquake-triggered landslides predominantly based on topography and PGV, and demonstrated that the addition of ICFs could improve the performance of these models.

Landslides can also be triggered by rainfall events, and as with earthquake-triggered landslides, cloud cover can cause significant problems with mapping landslides using optical satellite imagery. In particular, in areas with a monsoon climate (e.g. Nepal) the rainfall that triggers landslides often occurs during a period of several months with almost continuous cloud cover (Robinson et al., 2019). This means that it is currently extremely difficult to obtain landslide information beyond that predicted by static empirical models such as that of Kirschbaum and Stanley (2018) or the necessarily patchy and incomplete information

provided by reporting of individual landslides by individuals or organisations (e.g. Froude and Petley, 2018).

Landslides in vegetated regions are likely to result in decreased coherence regardless of their triggering mechanism due to associated movement of material and vegetation removal, which alters the microwave scattering properties of the Earth's surface. Therefore, it seems likely that the added prediction ability that can be gained by adding ICFs to empirical models of earthquake-triggered landslide susceptibility could also apply to models of rainfall-triggered landslide susceptibility.

### 4.8 Possible application in arid environments

The training and testing of our model was limited to landslides in vegetated areas. Due to significant differences in background InSAR coherence between arid and vegetated regions, it is likely that separate models would be required for vegetated and arid regions of the world. C-band InSAR is generally more successful in less heavily vegetated areas. Sentinel-1 ICFs might therefore play a larger role in empirical models based in arid environments than it did here. Sentinel-1 InSAR coherence has

been used to map erosion in arid environments (Cabré et al., 2020), indicating that it could also be applied to landslides. However, it is also more difficult to derive landslide inventories from optical satellite images in less vegetated areas. This makes it more difficult to obtain the landslide data required to train an empirical model, but it would also make such a model more valuable if it could be developed.

## 5 Conclusions

We have tested the relative performance of InSAR coherence based classifiers, empirical landslide susceptibility models and a combination of these using ROC analysis and $r^2$. The performance of all models was better with ROC analysis than with $r^2$, indicating that the models are better suited to discriminating between landslide and non-landslide areas following an earthquake than predicting continuous landslide areal density. We tested same-event and preliminary global empirical models and found that adding InSAR coherence features to these improved their performance in terms of both ROC and $r^2$. Importantly, a con-

siderable improvement in model performance was seen using SAR data acquired within 2 weeks of each earthquake, meaning



that these improvements could be made rapidly enough to be used in emergency response. We also expect that similar models could be developed to combine InSAR coherence and empirical models to predict rainfall-triggered landslides, or to predict earthquake-triggered landslides in more arid environments, although we did not test these cases here. For the same-event models, we observed that the empirical models without SAR performed considerably worse when the area available for training data was restricted as it could be by cloud cover following an earthquake, consistent with Robinson et al. (2017). However, we also found that the improvement in terms of ROC AUC offered by the inclusion of SAR data was particularly marked in this case. Both Sentinel-1 and ALOS-2 InSAR coherence features were observed to improve these models, with the best overall performance observed when both were used together. Before the acquisition of the first ALOS-2 image, when only Sentinel-1 InSAR coherence features were available, our same-event models outperformed our global models but after this our global models performed best. Our global models were only trained on two events, but the addition of the first ALOS-2 SAR data acquired after each earthquake resulted in improved and more consistent performance compared to the model of Nowicki Jessee et al. (2018) in terms of ROC AUC. We therefore recommend that in the future, InSAR coherence features from L-band SAR should be routinely incorporated into empirical models of earthquake-triggered landsliding in vegetated regions.

*Author contributions.* KB carried out data curation, investigation and formal analysis of the data and visualisation and writing of the original draft. Conceptualization and supervision of the work were carried out by RJW, DM and DB. Administration and funding acquisition were carried out by RJW and DM. All authors were involved in reviewing and editing the manuscript and in the methodology.

*Competing interests.* The authors declare no conflict of interest.

*Acknowledgements.* Sentinel-1 interferograms and coherence maps are a derived work of Copernicus data, subject to ESA use and distribution conditions. The authors are grateful to JAXA for providing ALOS-2 data sets under the research contract of RA-6 (PI No. 3228). Some figures were made using the public domain Generic Mapping Tools (Wessel and Smith, 1998). The work presented here was carried out as part of a PhD project funded by the Institute of Hazard, Risk and Resilience, Durham University, through an Action on Natural Disasters scholarship. This work was partly supported by the UK Natural Environmental Research Council (NERC) through the Centre for the Observation and Modelling of Earthquakes, Volcanoes and Tectonics (COMET). We thank Alex Densmore, Nick Rosser and Sang-Ho Yun for useful discussions and feedback on this work.



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


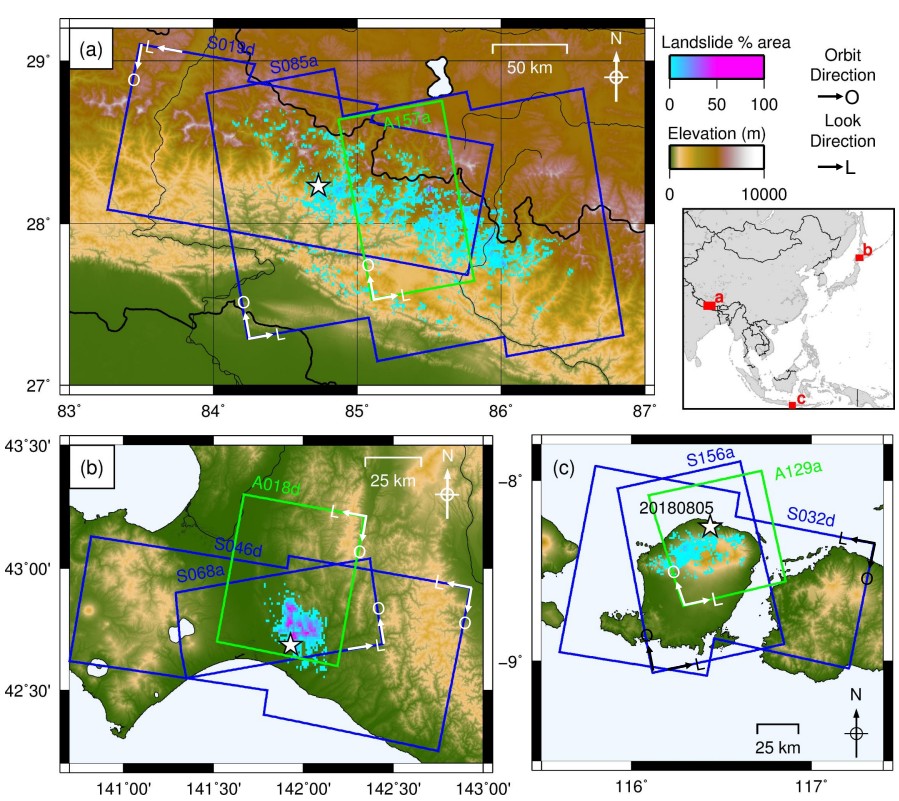

**Figure 1.** SAR coverage of the three case study regions: (a) the 2015 Gorkha, Nepal earthquake; (b) the 2018 Hokkaido, Japan earthquake; (c) the 2018 Lombok, Indonesia earthquake. Sentinel-1 scenes shown in blue. ALOS-2 scenes shown in green. Landslide data from Ferrario (2019); Roback et al. (2018); Zhang et al. (2019). White stars show earthquake epicentres. White arrows show satellite orbit (O) and look direction (L). Adapted from Burrows et al. (2020).

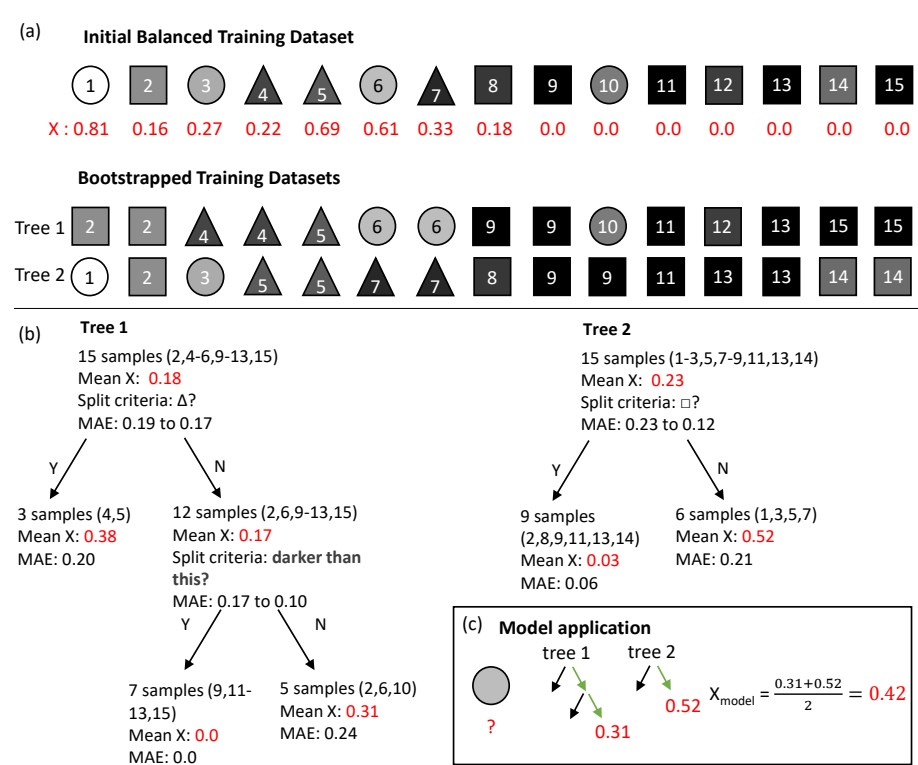

**Figure 2.** A simplified example of Random Forest Regression with a forest of two trees. The value of X for a sample is modelled based on its shape and colour. An initial dataset in (a) is bootstrapped to produce two slightly different datasets for use in tree 1 and tree 2. (b) Two trees which reduce mean absolute error. At each node, the number of samples, the mean value of the samples, the input feature on which the samples are split, and the mean absolute error of the samples before and after ("before - after") the split if they are assigned the mean value that is listed. In (c) the value of X for a new sample is modelled from the average output across both trees.

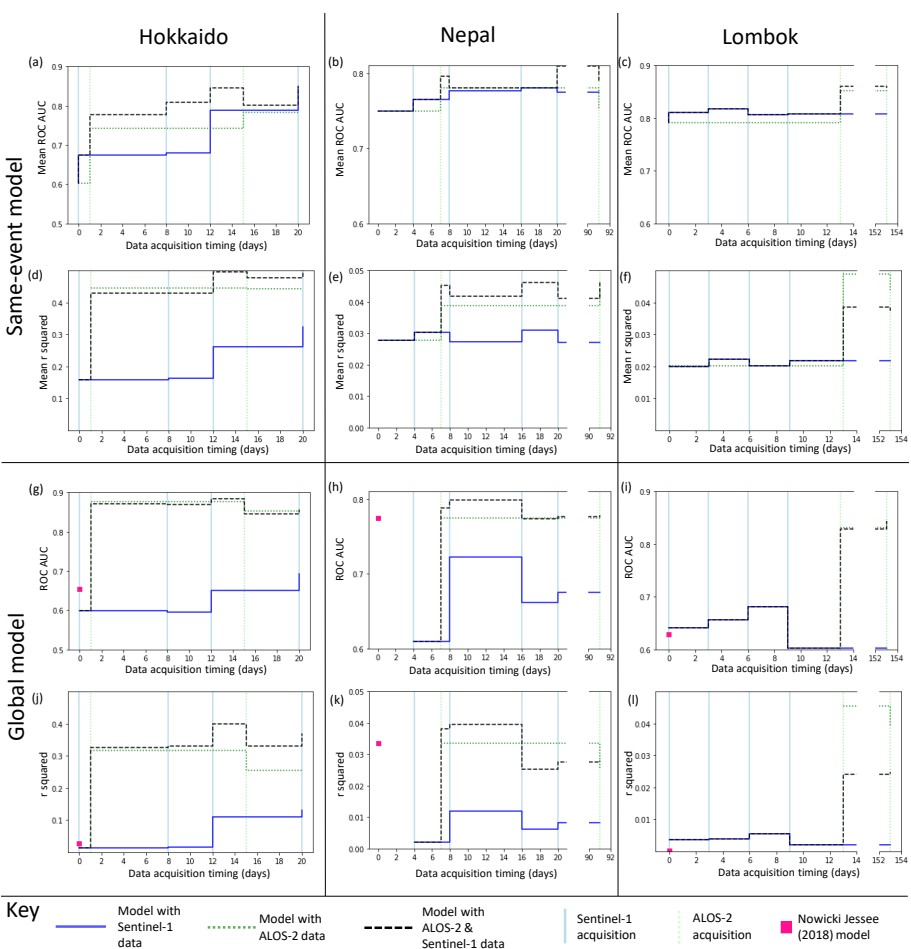

**Figure 3.** Step plots showing the change in ROC AUC (a-c, g-i) and $r^2$ (d-f, j-l) for each of the three events. Models in panels (a-f) were trained on a small part of each study area, while in panels (g-l) landslide areal density was predicted using a model trained on the other two case study regions in each case.

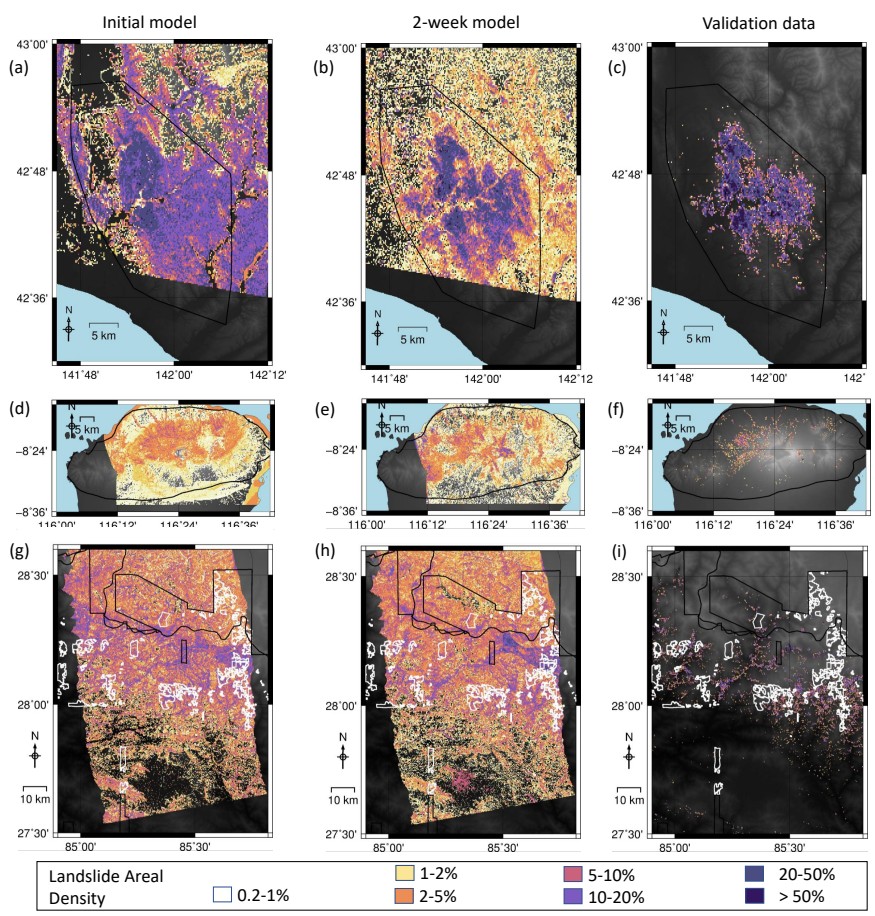

**Figure 4.** Timelines of modelled landslide areal density for each event, starting with no SAR data (left column). The second column shows the model using SAR data acquired within 2 weeks of the earthquake, and the final column shows landslide areal density calculated from polygon inventories for each event (Ferrario, 2019; Roback et al., 2018; Zhang et al., 2019). The mapping extent of each of these inventories is shown by a black polygon. White polygons in the final row show areas unmapped by (Roback et al., 2018) due to cloud cover in optical imagery.

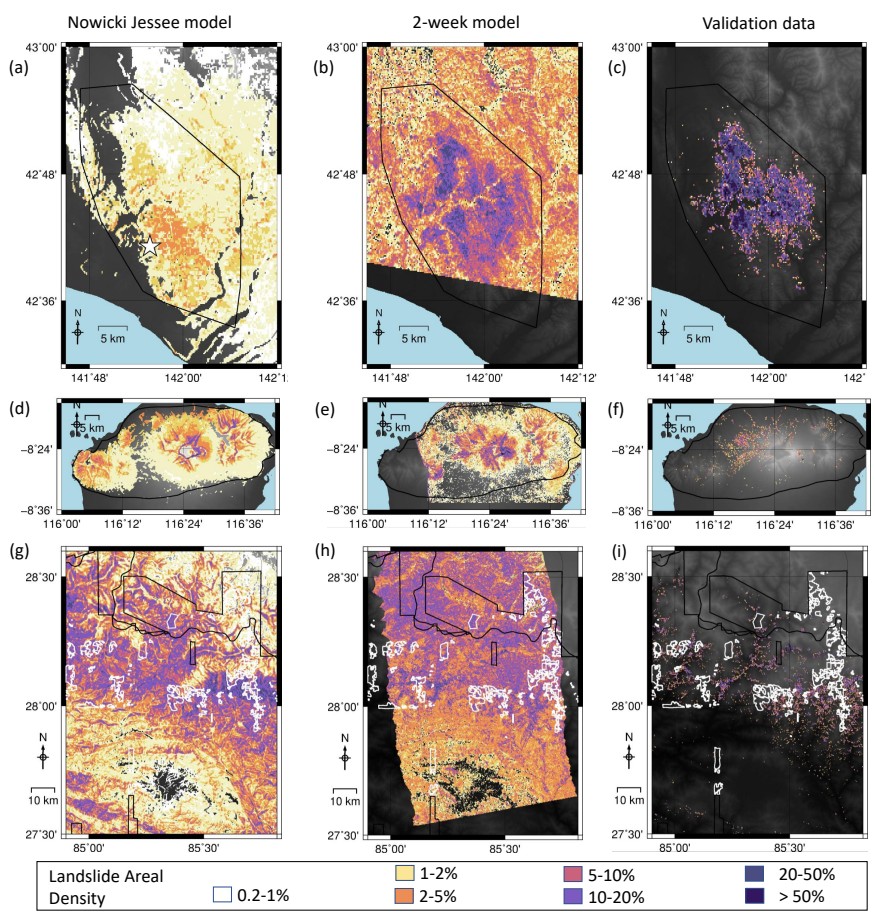

**Figure 5.** Timelines of modelled landslide areal density for each event. The first column shows the results of the model of Nowicki Jessee et al. (2018) for each event. The second column shows the model using SAR data acquired within 2 weeks of the earthquake, and the final column shows landslide areal density calculated from polygon inventories for each event (Ferrario, 2019; Roback et al., 2018; Zhang et al., 2019). The mapping extent of each of these inventories is shown by a black polygon. White polygons in the final row show areas not mapped by (Roback et al., 2018) due to cloud cover in optical imagery.

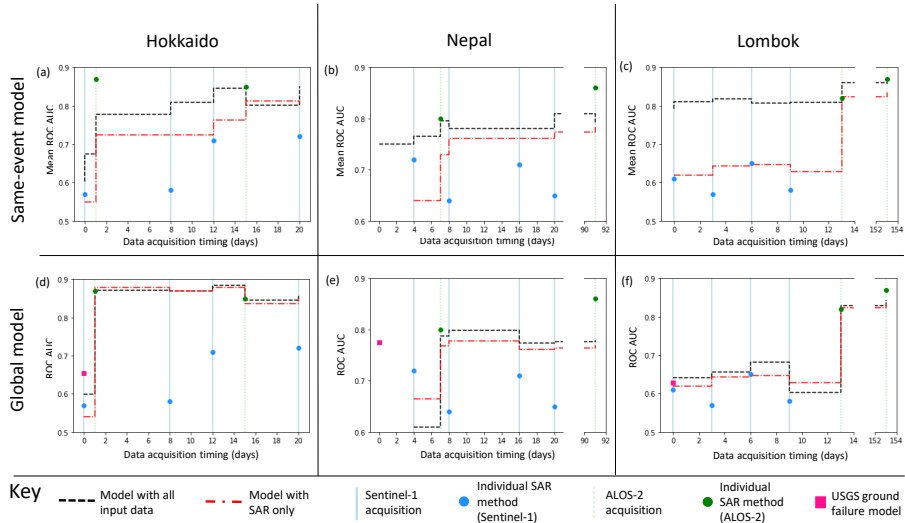

**Figure 6.** A comparison between the performance in terms of ROC AUC of individual SAR methods (blue and green circles), the model of Nowicki Jessee et al. (2018) (pink square), a combined model using all input data listed in Section 2.3 (black line) and a model using only SAR data (both Sentinel-1 and ALOS-2, red line)

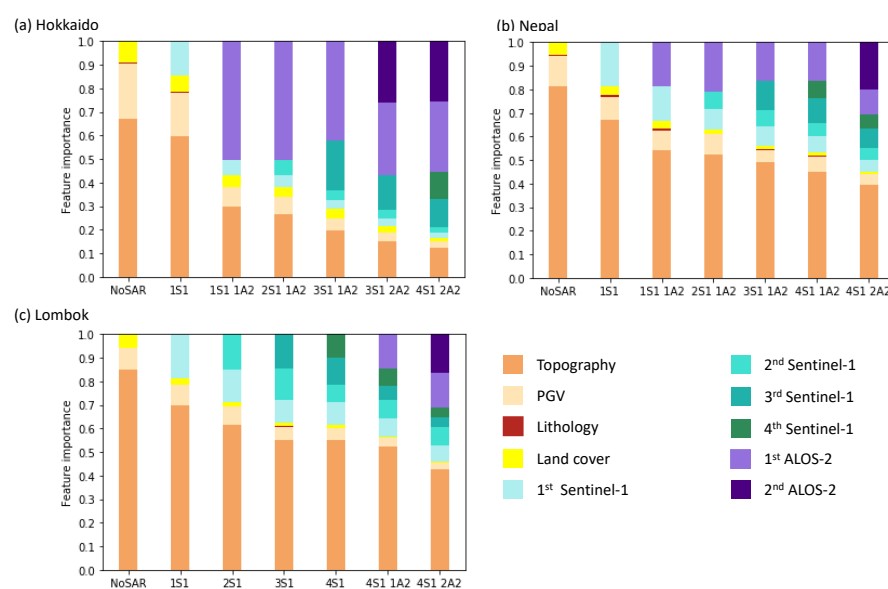

**Figure 7.** The relative importance of the different input datasets as ICFs generated from Sentinel-1 (S1) and ALOS-2 (A2) SAR data are added to the model for models trained on balanced landslide data mapped following (a) the 2018 Hokkaido, Japan earthquake, (b) the 2015 Gorkha, Nepal earthquake and (c) the 2018 Lombok, Indonesia earthquake

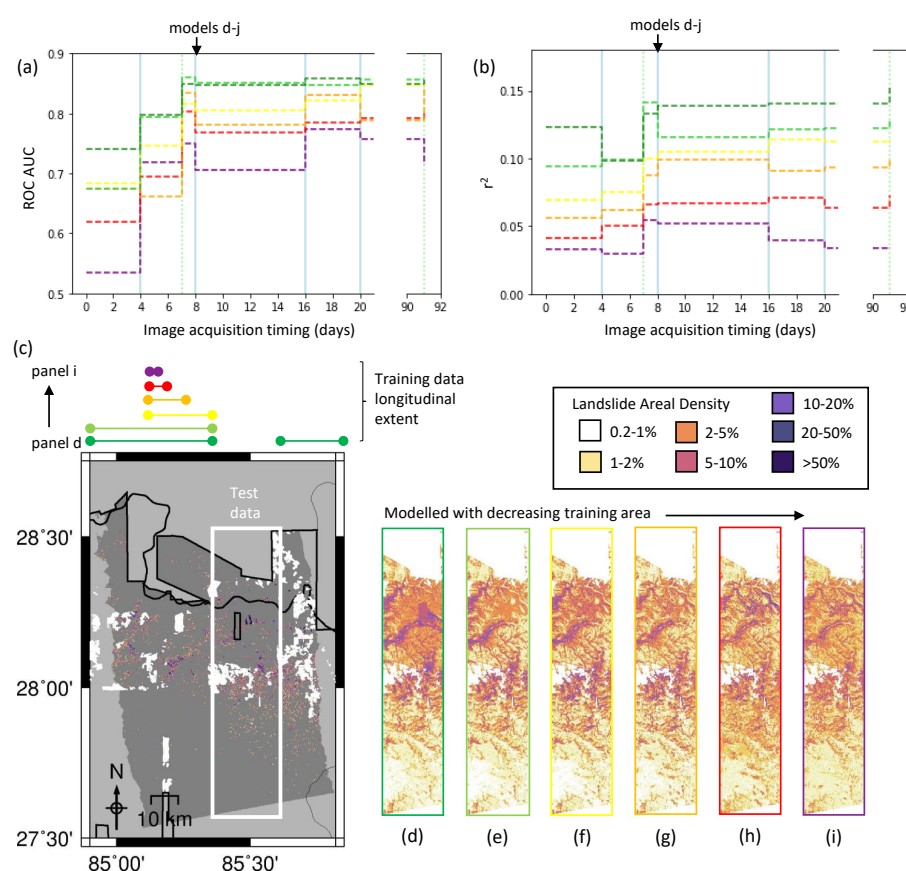

**Figure 8.** The effect of altering the extent of the training data in terms of ROC AUC (a) and $r^2$ (b). The colours of the lines in (a,b) correspond to the lateral extent of the training data marked on (c). (c) The mapped landslide inventory for the 2015 Gorkha earthquake of Roback et al. (2018), reproduced as landslide areal density for 200 x 220 m pixels. (d-i) Modelled landslide areal density with decreasing areas of data used for training the model.





**Table 1.** Input Features used in the Random Forest models and whether they were used in the same-event or global models.The table is divided into conventional predictors of landslide likelihood and and those based on InSAR coherence. For the latter, the case where only data from the Sentinel-1 (S-1) satellite were used is noted, otherwise both Sentinel-1 and ALOS-2 data were used.

| Input feature | Same-event models | Global models |
|---|---|---|
| Mean elevation | ✓ | |
| Standard deviation of elevations | ✓ | |
| Maximum Slope | ✓ | |
| Mean Slope | ✓ | |
| Standard deviation of slopes | ✓ | |
| Circular mean of slope aspect | ✓ | |
| Mean compound topographic index | ✓ | |
| Relief | ✓ | |
| Peak ground velocity | ✓ | |
| Lithology | ✓ | |
| Land cover | ✓ | |
| USGS ground failure product | | ✓ |
| Co-event coherence | ✓ | ✓ |
| Co-event coherence loss (CECL) | ✓ | ✓ |
| Boxcar-Sibling coherence (Bx-S) | ✓ (S-1 only in Nepal) | ✓ (S-1 only) |
| Post-event coherence | ✓ | ✓ (S-1 only) |
| Post-event coherence increase (PECI) | ✓ | ✓ |
| Sum of CECL and PECI | ✓ | ✓ |
| Maximum CECL or PECI | ✓ | ✓ |




**Table 2.** Hyperparameter options for the Random Forest in Scikit-learn over which a grid search optimisation was carried out (Pedregosa et al., 2011).

| Hyperparameter | Definition | GridSearch |
|---|---|---|
| n_estimators | The number of decision trees that make up a forest | [75, 100, 125] |
| max_features | The number of input features (as a function of the total) considered when looking for a split | ["log2","sqrt"] |
| max_depth | The maximum depth of the tree | [10,15,20,30] |
| min_samples_split | The minimum number of samples at a node for a split to be allowed | [2,3,4,5] |
| min_samples_leaf | The minimum number of samples that would result at each leaf for a split to be allowed | [2,10] |