# Peer review of "Integrating empirical models and satellite radar can improve landslide detection for emergency response"

_Natural Hazards and Earth System Sciences, 2021_

## Author Response (AR1)

**Author's response to comments**
We thank the reviewers for their comments, which have helped to improve the quality and clarity of the manuscript. Here we provide a compete response to comments from both reviewers and from the editor, along with the revised manuscript in which all changes are indicated

Reviewer 1
**Comment 1**
Data re-sampling consequences seem to be underestimated or not fully considered: re-sampling is here used to overlap layers with different resolutions but, when data are re-sampled, at least two points/questions must be taken into account: (I) is my process still represented/sampled at the correct scale? (II) Does re-sampling introduce artifacts (e.g. aliasing) that in quantitative analysis can impact the results?
Apart from the re-sampling, the original scales/resolutions of the data are not reported but they look to be quite different. Are they all adequate to sample the different processes and to be combined in a machine learning model (no ghost relationships)?
**Response**
We address this point in more detail in response to the reviewer's specific questions. It is true that by coarsening the resolution of our input products, we have lost information. However, the resolution at which we have applied our model is similar to that used by other studies including the landslide model of Nowicki Jessee et al. (2018) that is published online by the USGS following an earthquake for use in emergency response.
**Manuscript Change**
For changes to the manuscript, see Reviewer 1 Comments 10,19, 21 and 24

**Comment 2**
The results are (quite well) numerically discussed but the geomorphological part is missing. So we know where and when there are numerical advantages in the combination of the two products, but we don't know why (e.g. geomorphological characterization of the true/false positives/negatives), this makes it difficult to evaluate the real possibility to export the framework in different contexts (e.g. Par 4.4)
**Response**
We believe that the geomorphic implications of the models are beyond the scope of the paper, which is aimed at application during an emergency response. However, this is an excellent suggestion for further work on the topic, and would be appropriate to include in a follow-on paper aimed at wider evaluation of our approach across a larger global dataset of events.
**Manuscript Change**
No changes made to main body of text, but the title now contains the words "emergency response" to better represent the paper's aim.

**Comment 3**
There is a point that I admit, I still have problems to unravel. I feel that the input variable ICF is somehow a proxy of what the models want to find (the output variable). Furthermore, it sounds to me as if the ground truth was used twice: the first time when landslides (Y - independent variable) are exploited to label the units, and then, when the ICFs say (again, but as feature variable - X) to the model where landslides occurred. Can a regression/classification model make use of variables that somehow are what the model wants to predict/classify (no spurious correlations?)? If yes, can their importance be evaluated in the same way the importance of the other variables is evaluated? My doubt can be originated by the difficulties I'm having to interpret the output of the model as landslide mapping instead of landslide (spatial) forecasting (see my several comments later.
**Response**

We have responded to the individual comments later in the text. But the key point is that the ICFs are indeed a proxy for landslide distribution (as mapped from optical imagery), just as the various other model inputs are (whether they are static such as topographic slope, or dynamic such as peak ground velocity). And just as the other model inputs have limitations and are imperfect proxies/predictors, so do the ICFs – i.e. while InSAR coherence is sensitive to landslides, it is not purely a measure of landslides, since other changes to the ground surface e.g. building damage, soil moisture changes, vegetation growth can result in the same signal as a landslide. So essentially, what we aim to predict is the landslide distribution (as mapped from optical satellite imagery), from a number of imperfect proxies. Therefore, of course we hope that the ICFs and optical maps are correlated, just as we hope that PGV and the optical maps are correlated. But neither of these are spurious correlations because they are comparisons between imperfect predictions and independent 'ground truth' of landslide distribution obtained from optical datasets.

See responses to Reviewer 1 Comments 29, 36, 41 for more detail and manuscript changes.

**Comment 4**
It would also be helpful to evaluate how ICFs false positive and negative (random, systematic, reproducible?) impact the model.
**Response**
We observed that false positives that were present in the InSAR coherence products of Burrows et al (2020) that were attributed to vegetation damage or to built-up areas are dampened in the models we present here.
**Manuscript Change**
New text: "*We are also able to make a visual comparison between the InSAR coherence products of Burrows et al. (2020) and the 2-week models shown here in Figures 3 and 4. Burrows et al. (2020) observed false positives south of the landslides triggered by the Hokkaido earthquake, which they attribute to wind damage to vegetation associated with Typhoon Jebi. They also observed false positives in built-up areas close to the coast in Lombok. These areas of false positives are visibly dampened in both our same-event and our global 2-week models.*" (Lines 406-410 of the revised manuscript)

**Comment 5**
Title: not so sure the title is correct: the empirical models don't detect landslides but they model the spatial probability of occurrence.
**Response**
The choice of word is tricky since we are combining a method of landslide detection with a method of landslide prediction. InSAR coherence can be used in landslide detection but the empirical model is a prediction of where landslides are most likely to have happened. Based on this comment, we considered changing the title to *"Integrating empirical models and satellite radar improves landslide prediction for emergency response*". Following further suggestions from the editor, we have changed it to ""*Integrating empirical models and satellite radar can improve landslide detection for emergency response"* (See Editor Comment 1).
**New Text**
Original manuscript title "*Improved rapid landslide detection from integration of empirical models and satellite radar*"
New manuscript title "*Integrating empirical models and satellite radar can improve landslide detection for emergency response*"

**Comment 6**
20-25: "generating information on the intensity and spatial the extent of landslides in the immediate aftermath of a large earthquake": what does intensity mean here?

**Response**

By intensity, we meant the area affected by landslides, but since this is covered by "spatial extent", we will simplify by removing "intensity and" from the sentence.

**Manuscript Change**

"*there are two options for generating information on the intensity and spatial extent of triggered landsliding in the immediate aftermath of a large earthquake*" at lines 23-24 of the original manuscript has been changed to "*there are two options for generating information on the spatial extent of triggered landsliding in the immediate aftermath of a large earthquake*" at lines 22-23 of the revised manuscript.

**Comment 7**

25-44: The discussion about whether to consider driving factors or not into susceptibility maps is quite hot and open. I suggest anticipating here the comment at par 2.3.2 to avoid possible misunderstandings.

**Response**

In this paragraph, we are discussing methods of generating landslide information immediately after an event. Empirical models that do not incorporate causative factors do not fit into this category since they are not event specific. To make it clearer the type and application of empirical model we are talking about here, we make the following change:

**Manuscript Change**

"*To generate an empirical model of triggered landslides following an earthquake, a training dataset of mapped landslides is analysed alongside maps of factors known to influence landslide likelihood such as slope, land cover and ground shaking estimates, and a model is produced that predicts landslide likelihood based on these inputs.*" at lines 31-33 of the original manuscript has been changed to "*To generate an empirical model of triggered landslides following an earthquake, a training dataset of landslides is analysed alongside maps of 'static' factors known to influence landslide likelihood e.g. slope and landcover, as well as 'dynamic' causative factors e.g. ground shaking estimates, and a model is produced that predicts landslide likelihood based on these inputs.*" (lines 29-32 of revised manuscript)

**Comment 8**

45: maybe helpful: https://doi.org/10.1016/j.earscirev.2021.103574

**Response**

Thank you, this paper is highly relevant and has been added to the reference list.

**Manuscript Change**

This paper is now referenced at lines 28 and 550 of the revised manuscript. It has been added to reference list at line 713

**Comment 9**

60-65 'Here we aim to establish which of these three options is best': susceptibility and detection provide different types of results (LAD probability, and LAD measure), and what is best is not absolute but it depends on how they are used. I suggest explaining what 'best' means here.

**Response**

As the paper is oriented towards emergency response, we have assumed that the three methods will all be used in the same way: to indicate areas which have been heavily damaged by triggered landsliding following an earthquake. To make this clearer, we make the following change:

**Manuscript Change**

Changed "*which of these three options is best*" at line 63 of the original manuscript to "*which of these three options provides the best indication of areas strongly affected by triggered landslides after an earthquake*" at lines 62-63 of the revised manuscript.

**Comment 10**

70-75 "We chose to model LAD rather than individual landslide locations as both empirical models and SAR-based methods perform best at relatively coarse spatial resolutions": It does not sound very clear. I suggest rephrasing the sentence and try to explicit more the relationship between spatial resolution and the models (I guess susceptibility and LAD from coherence and not random forest). Another possibility can be to demand this concept to the discussion.

**Response**

Yes, since SAR methods which use polarimetric data, or those which are based on SAR backscatter are often applied at a higher resolution to detect individual landslides, this sentence was too vague. We have rewritten it for improved clarity, and also note that the improvement with coarser resolution is supported by previous work on the ICFs. Empirical models are usually generated at the resolution of their coarsest input feature, which requires them to model landslide intensity rather than individual landslides.

**Manuscript Change**

We have changed "*We chose to model LAD rather than individual landslide locations as both empirical models and SAR-based methods perform best at relatively coarse spatial resolutions (within the range 0.01-1 km$^2$ (Burrows et al., 2019; Nowicki Jessee et al., 2018; Robinson et al., 2017)).*" in the original manuscript to "*Rather than attempting to delineate individual landslides, we chose to model LAD as both empirical models and detection methods based on InSAR coherence perform well at relatively coarse spatial resolutions (within the range 0.01-1 km$^2$ (Burrows et al., 2019; Nowicki Jessee et al., 2018; Robinson et al., 2017)).*" (lines 70-73 of the revised manuscript)

**Comment 11**

109-115: 'a small number of landslides', small, but I guess the number must be representative from a statistical point of view and a geomorphological point of view, in the sense that it should be statistically significant and well representing the density of landslides in the area, correct?

**Response**

This is correct. The number of mapped landslides can be small but the samples must be dispersed throughout the affected area in order to provide a representative sample of the affected area

**Manuscript Change**

Changed "*The distribution of these landslides can then be used to predict the landslide distribution across the whole affected area…*" at lines 110-111 of the original manuscript to "*Providing these landslides are dispersed throughout the study area to constitute a representative training dataset, their distribution can then be used to predict the landslide distribution across the whole affected area…*" at lines 110-112 of the revised manuscript.

**Comment 12**

'we randomly selected 250 landslides…': if I understand well, you labeled as yes, those pixels in which LAD > 0.01, and no elsewhere, correct?

**Manuscript Change**

To improve clarity, we have changed "*we randomly selected 250 landslide (LAD > 0.01) and non-landslide pixels*" at lines 112-113 of the original manuscript to "*we randomly selected 250 landslide (LAD >=0.01) and 250 non-landslide (LAD > 0.01) pixels for use as training data.*" at line 113 of the revised manuscript.

**Comment 13**

115-120 'we randomly undersampled': I guess you mean that you randomly selected the pixels, correct?

**Manuscript Change**

To improve clarity, we have changed "*we randomly undersampled both the landslide (LAD > 0.01) and non-landslide pixels to select 1000 landslide and 1000 non-landslide from each event*" at lines

117-118 of the original manuscript to "*we randomly selected 1000 landslide (LAD >= 0.01) and 1000 non-landslide (LAD < 0.01) from each event*." at lines 117-118 of the revised manuscript

**Comment 14**

115-120 'resulting model was therefore trained on equal numbers of pixels from the two training events': interesting point. The choice is probably dictated by the model, I'm wondering what implications this choice can bring. Don't you 'de-correlate' the strength of the earthquake (represented somehow by the ground shaking data) and the number of landslides (the consequence of the strength) (and other covariates)? Forgive me if I'm using some terms inappropriately.

**Response**

It is true that a larger earthquake will generally trigger more landslides, but it is not necessarily the case that the larger earthquake will have higher LAD values. For example, out of the cases we look at here, the Hokkaido earthquake is Mw 6.6, while Nepal is Mw 7.8, and indeed the Nepal earthquake triggered many more landslides across a much larger area, but Hokkaido has more cells with high LAD values than Nepal.

Furthermore, if we used more samples from large earthquakes compared to small earthquakes, the model could be biased to recreate landsliding more similar to one event than the other. This is particularly important here, since we are only using two events as training data. We therefore think it better to use equal numbers of landslides from each event.

**Manuscript Change**

We have changed "*The resulting model was therefore trained on equal numbers of pixels from the two training events, rather than being dominated by whichever was larger*" at lines 118-119 of the original manuscript to "*The resulting model was therefore trained on equal numbers of pixels from the two training events, which prevents it from being dominated by the larger training event*." at lines 118-119 of the revised manuscript.

**Comment 15**

120-125 'each model within the ensemble is trained on a different set of cells': do you estimate/suppose that the inventories have locally the same quality?

**Response**

We do not expect all landslide inventories to be of equal quality. It is for this reason that we repeat the sampling process 100 times and take the median of the ensemble prediction. This should reduce the influence of any single training dataset that could be of poorer quality by chance (i.e. one that is not representative or is biased in some way). Using different sets of cells allows us to produce a more diverse ensemble, resulting in a more robust prediction.

Coarsening the resolution from individual landslide polygons to LAD should also help with this, since it should overcome differences in e.g. detail (the number of points used to delineate the landslide) or how precisely the landslides are located.

**Manuscript Change**

None

**Comment 16**

120-125 'This process reduces variability': in what sense?

**Response**

It is necessary to take equal numbers of landslide and non-landslide pixels for every event, which requires us to undersample the data, but exactly which pixels are selected affects the model output. By 'variability' we meant that if you run the model several times, its prediction and performance changes slightly each time because it is using different sets of pixels. However if you run the ensemble of models several times, the median output does not vary as much. This was not very clear in the text, we have updated this.

**Manuscript Change**

We have changed "*This process reduces variability*" at line 122 of the original manuscript to "*This process allows the model to use more of the available training data and improves robustness*" at line 122 of the revised manuscript

**Comment 17**
125-150 'Masato...': I suggest to better contextualise: the model tuned using rainfall does not use (a polite guess...) shaking data...
**Response**
Masato model uses only SAR and topography, it does not use any causative input. We have altered the text to clarify this.
**Manuscript Change**
We have changed "*Masato et al. (2020) demonstrated that when using fully polarimetric SAR data, it was possible*" at lines 126-127 of the original manuscript to "*Ohki et al (2020) demonstrated that when using fully polarimetric SAR data alongside topographic input features, it was possible…*" at lines 127-128 of the revised manuscript
(Reviewer 2 (Comment 3) indicated that Ohki, rather than Masato was the family name of the author – we are referring to the same paper here as before)

**Comment 18**
130-135 'Instead, a high performance on at least one predicted event would be considered a success': I would be a bit more cautious, one result is not statistically significant, I suggest to cancel the sentence and say (correctly) that would encourage further investigation.
**Manuscript Change**
We have changed "*Instead, a high performance on at least one predicted event would be considered a success; this would suggest that such an approach is worthy of further investigation and is likely to improve when trained on a number of case-studies more comparable to those used by current "gold-standard" models*" at lines 131-134 of the original manuscript to "*Instead a high performance on at least one predicted event would suggest that our approach is worthy of further investigation and is likely to improve when trained on a number of case-studies more comparable to those used by current "gold-standard" models*" at lines 132-134 of the revised manuscript.

**Comment 19**
2.3 Input Features
Two general comments related to the fact that this is a quantitative (and not qualitative or based on interpretation) analysis:
1) resampling to a higher resolution can be problematic, what type of resampling did you use?
2) how did you aggregate here? In this case, have you considered aliasing problems? Is the result still sampling the variable at the right scale?
**Response**
The resolution of these input data was finer than or similar to the resolution of our model, so we did not resample to a higher resolution. For our resampling, we use linear interpolation for continuous input features (e.g. slope) and nearest-neighbour for categoric input features (e.g. lithology). Information on resampling, aggregation and on the resolution of the original data is now included for each input feature.
It is true that perhaps small-scale features are lost as we coarsen the resolution, but there is no way to avoid this, since we need to obtain a set of input features in the same geometry that are derived from different datasets with different spatial resolutions. For topography, the number of different derived surfaces goes some way to address this. For example, if the mean slope of a cell does not represent the fact that a small area of the aggregate cell is very steep while the rest is flat, this will be picked up by the standard deviation of slopes within the aggregate cell, or by the maximum slope within the aggregate cell.
**Manuscript Change**

Where this information was missing, we have added information on the resolution of the input data. See response to Reviewer 1 Comments 21 and 24.

**Comment 20**
150-155 'a static proxy for soil moisture': did you find it relevant (from a geomorphological point of view) for earthquake-induced landslides? Or just from a numerical point of view?
**Response**
The proxy is used by Nowicki Jessee (2018) and Tanyas et al. (2019). It is relevant because it is a metric for spatial variation in phreatic surface depth and thus pore pressure (e.g. Montgomery and Dietrich, 1994), which in turn influences material strength (and thus stability) by reducing effective normal stress on the failure surface.
**Manuscript Change**
We have changed "*the compound topographic index (CTI), which provides a static proxy for soil moisture (Moore et al. 1991)*" at line 151 of the original manuscript to "*the compound topographic index (CTI), a static proxy for pore water pressure, which alters the effective normal stress on the failure surface and thus the material strength (Moore et al. 1991)."* at lines 152-153 of the revised manuscript.

**Comment 21**
2.3.2 Ground shaking estimates
General comment: I suggest adding the scale/resolution of the map, and eventually its uncertainty. Since this layer is also quite different from the others (sort of causing factor) I also suggest commenting (maybe not here) how this can be an adequate sampling of the measure in relation to the product you want to obtain and its resolution. Is the map enough resolute to influence/characterise the LAD? Is it compatible with the other products?
**Response**
The grid spacing is 0.00083 degrees (~92 m at equator). We will add this information to the text. This is a similar resolution (slightly finer) than the resolution of our model (~200 m) and so we expect that the product should be well suited to characterising LAD. The USGS ground failure products, which are generated at a very similar resolution to the models we present here, also make use of this shaking intensity product and find it is well-suited to landslide hazard modelling
With regards to the uncertainty of the ground shaking models, this varies depending on the local seismic network and is not currently incorporated into empirical models so we do not consider it here.
**Manuscript Change**
"*Gridded PGV estimates are available with pixel spacing 0.00083∘ as part of the USGS ShakeMap product. We resampled this onto the 200 x 220 m geometry used here using a linear interpolation. The uncertainty of the ShakeMap product varies depending on the local seismic network used to gather the shaking data on which the shaking estimates are based (Wald et al. 2020) but is not currently taken into account when using these estimates in empirical models (Nowicki Jessee et al, 2018)."* added at lines 165-169 of the revised manuscript

**Comment 22**
160-165 'is what distinguishes': so, is there any way to call it differently?
**Manuscript change**
We have changed "*is what distinguishes*" at line 161 of the original manuscript to "*is what differentiates*" at line 161 of the revised manuscript.

**Comment 23**
170-176 'Our inital susceptibility model may therefore perform better than one that uses the data made available within the first few hours of the earthquake...study': this should be part of the

discussion. In any case, I still strongly recommend testing and compare the two products, since you are evaluating a progressive improvement of the performances of the system devoted to working in an emergency phase, and SAR images might be available just after the event. (I guess inital is a typo).

**Response**

There are more than two products to consider here – the ShakeMap product is updated multiple times after an earthquake. We feel it would make the manuscript overly complicated to incorporate successive changes to the model that are a result of updates to the ground shaking estimates as well as the changes we can obtain by incorporating ICFs. We have rewritten this paragraph to better explain why we use only the final ShakeMap product in our models.

**Manuscript Change**

"*An initial estimate of ground shaking is generally available within hours of an earthquake from the USGS ShakeMap webpage and is then refined as more data become available. Allstadt et al (2018) demonstrated that in some cases, such as the 2016 $M_w$ 7.8 Kaikōura, New Zealand earthquake, the version of the shaking estimate used can have a significant impact on the modelled landsliding. This was also observed in the case of the $M_w$ 7.1 2018 Anchorage, Alaska earthquake (Thompson et al. 2020). Although our results are presented as a timeline where the model was revised as SAR data became available, we used the same version of PGV throughout, which is the final available model for each event. Our inital susceptibility model may therefore perform better than one that uses the data made available within the first few hours of the earthquake. However, the availability of SAR data for ground deformation measurements is one of the factors that can significantly improve the shaking estimate, so in practice, it will only be possible to add SAR-based landslide indicators to the final or close-to-final versions of these empirical models. It is also preferable to isolate the effect of incorporating SAR by keeping all other input features the same in this study*." at lines 166-177 of the original manuscript has been changed to

"*An initial estimate of ground shaking is generally available within hours of an earthquake from the USGS ShakeMap webpage and is then refined as more data become available. The difference that these updates to the ground shaking estimates can make to empirical models of landsliding has been explored by Allstadt et al. (2018) and Thompson et al. (2020). Here our aim was to investigate changes to modelled landsliding due to the incorporation of SAR data and so for simplicity we chose to keep all other input features constant through time. We used the final version of PGV published for each event in all our models. In reality, model performance would evolve both due to updates to the estimated PGV and the availability of SAR data. But SAR data are also used to improve PGV estimates, and the final 'best' PGV estimate is often achieved when the first post-event SAR data are incorporated. Therefore, in practice, SAR-based landslide indicators will only be added to final or close-to-final versions of these empirical models.*" at lines 171-178 of the revised manuscript.

**Comment 24**

**2.3.2 Lithology**

General comment: I suggest adding the scale/resolution of the map

**Response**

The GliM is provided as vector data, with a resolution that varies depending on exact location since it was generated through combining national geological maps from a range of sources. However, these data were used by Nowicki Jessee (2018) who found them suitable for generation of an empirical model at the same scale as ours.

**Manuscript Change**

"*For each 200 x 220 m aggregate pixel, we took the dominant basic lithological class…*" at line 182 of the original manuscript changed to

"*These data are provided in vector format. We rasterized these data onto the 20 x 22 m grid at which the SAR data were processed. Then for each 200 x 220 m aggregate pixel, we took the dominant basic lithological class…*" at lines 183-185 of the revised manuscript.

**Comment 25**

180-185 'One advantage of Random Forests is their ability…': I suggest moving this elsewhere

**Response**

We placed this here to justify this processing step, but will shorten it.

**Manuscript Change**

"*One advantage of Random Forests is their ability to accept both continuous and categoric input features*" at lines 183-184 of the original manuscript has been changed to "*Random Forests can accept both continuous and categoric input features*" at line 186 of the revised manuscript.

**Comment 26**

2.3.5 USGS ground failure products

General comment: I suggest removing/move the first part of the paragraph. Here only the method should be described and not the reasons for which others failed. In the discussion, you can say why you did not use the same model…

**Response**

We feel that the first paragraph of this section is necessary as it justifies using this product as an input feature for the global models instead of building them from the same input features as our same-event models. The paragraph has been rewritten as follows:

**Manuscript Change**

"*For our global model, in which two events were used as training data and the third was predicted, we found that it was not possible to generate a reasonable initial model using the landslide influencing factors listed in Sections 2.3.1-2.3.4. The models generated in this way had limited skill in Hokkaido and no skill in Lombok or Nepal, which is not representative of the performance of existing global models (e.g. Nowicki Jessee et al., 2018; Tanyas et al., 2019). This poor performance is likely to be due to the limited number of case studies used as training data in our global models and would make it difficult to draw conclusions on the benefits of adding SAR to a global empirical model. Therefore, instead of using the set of individual input feature maps in the global model, we used one single feature map that encapsulates the likely best results from a global model trained across a large number of events: the output USGS Ground Failure product of Nowicki et al. (2018).*" at lines 199-207 of the original manuscript changed to:

"*For our global model, instead of the set of individual input feature maps described above, we use one single feature map that encapsulates the likely best results from a global model trained across a large number of events: the output USGS Ground Failure product of Nowicki Jessee et al. (2018). This was necessary since we found that models generated from individual input feature maps had limited skill in Hokkaido and no skill in Lombok or Nepal, which is not representative of the performance of existing global models (e.g. Nowicki Jessee et al. 2018; Tanyas et al. 2019). This poor performance is likely to be due to the limited number of case studies used as training data in our global models and would make it difficult to draw conclusions on the benefits of adding SAR to a global empirical model.*" at lines 201-207 of the revised manuscript.

**Comment 27**

And again, I suggest finding another way to say that the model would have probably worked worse if you had used another product… this sounds like 'incomplete'.

**Manuscript Change**

We have changed *"these models evolve in the days and weeks following an earthquake as the best available PGV estimate is updated. Here we used the final version of each model, which may perform better than the model version that would be available within hours of an earthquake."* at lines 212-214 of the original manuscript to "*USGS estimates of PGV are updated multiple times after an earthquake and the ground failure products based on these are also updated. Again, we use final published version of the ground failure product (based on the final PGV estimates) in all our models.*" at lines 213-215 of the revised manuscript.

**Comment 28**

2.3.6 InSAR coherence features (ICFs)

230-235 'This volume of SAR data..' for sure for S1, also true for ALOS-2?

**Response**

In most cases, we expect at least one ALOS-2 image to be available within two weeks of an earthquake, since the satellite is tasked with acquiring imagery following an earthquake for emergency response and has a 14 day repeat time.

**Manuscript Change**

None

**Comment 29**

240 – 245 'Burrows et al (2020) … and 2019, and the whole paragraph: I had a look at the papers, unfortunately, I'm not sure I got correctly a point: the different methods are based on differences (generally speaking) and to decide what is the right threshold, a ROC analysis is required. A benchmark is then required, so you need to have the landslide map already prepared. I see a kind of loop. Can you please clarify? Or, If I'm wrong, can you please make explicit when 'lower' or higher' is low enough or high enough to say that the changes were caused by landslides?

**Response**

In the papers, a threshold was not applied, since the tolerance for false positives and false negatives is dependent on factors such as the available resources during the emergency response phase and this was beyond the scope of the papers. Therefore, the output from these papers is a continuous surface between 0 and 1 with 1 indicating a strong likelihood of landsliding and 0 indicating the opposite. ROC analysis was used to evaluate the different continuous outputs independent of the choice of threshold, not to choose a specific threshold. It is these continuous surfaces which are used as inputs in the model. This is clarified in the new version of the document.

**Manuscript Change**

Sentence added at line 258 of revised manuscript: "*The output of each of these methods is a continuous surface that can be interpreted as a proxy for landslide intensity*."

**Comment 30**

255 – 260 'showed some level of landslide predictive skill': not sure I would use predictive here, I suggest detection capacity or similar

**Manuscript Change**

"*landslide predictive skill*" at line 256 of original manuscript has been changed to "*landslide detection skill*" at line 258 of the revised manuscript.

**Comment 31**

2.4 Random forest theory and implementation

General comment: not sure that Fig 2 and some parts of the description of RF are really helpful. My suggestion is to remove fig 2, shorten the paragraph, and eventually add some references.

**Manuscript Change**

We have shortened this section by removing the figure and all references to it from the main body of the text. The figure is now included in the supplementary material. Subsequent figures have been automatically renumbered in the text and figure captions by LaTeX.

Removed "*A very simple example of the Random Forest method using only two trees to estimate the value a sample should be assigned based on two input features (the colour and shape of each sample) is shown in Figure 2*", lines 277-279 of original manuscript

Added "*To aid understanding, we include a simple example of Random Forest Regression in the supplementary material*" at line 285 of the revised manuscript

Removed references to Figure 2 at lines 280, 281, 283, 296 of the original manuscript.

Changed "*Max_depth defines the maximum "depth" of each decision tree i.e. the longest path length from the beginning of the tree to the end. For example in Figure 2, Tree 1 has a depth of 3 and Tree 2 has a depth of 2*" at lines 305-307 of the original manuscript to "*The "depth" of each decision tree describes the number of times the data will be split if it takes the longest path from the beginning of the tree to the end. This is limited by Max_depth, which defines the maximum "depth" of each decision tree.*" at lines 307-309 of the revised manuscript.

**Comment 32**
325 – 330 'by comparing the test areas of these predicted surfaces with the mapped LAD calculated in Section 2.1.': LAD obtained from the external inventories, correct?
**Response** That is correct, we have changed the text to improve clarity.
**Manuscript Change**
We have changed "*We assessed model performance by comparing the test areas of these predicted surfaces with the mapped LAD calculated in Section 2.1*" at lines 235-236 of the original manuscript to "*We assessed model performance by comparing the test areas of these predicted surfaces with the mapped LAD calculated from the inventories of Ferrario (2019); Roback et al. (2018) and Zhang et al. (2019) in Section 2.1*" at lines 326-329 of the revised manuscript.

**Comment 33**
330 – 335: I understand the need of choosing a threshold, in fact, I think the most appropriate sentence to define what is tested here is "The ROC AUC values calculated here, therefore, represent the ability of the model to identify pixels with LAD > 0.1" but, according to the fact that the percentages are so different for Hokkaido and for Lombok, I'm not so sure that the value can have the same weight in the testing phase (different densities related to different events/geosettings).
**Response**
It is true that the proportion of landslides from the three optical satellite images that are represented by this 0.1 threshold is different. However, in general for emergency response it is more important to detect bigger landslides or areas of more intense landsliding, and so we choose to keep this value of 0.1 constant. The proportion of small landslides mapped in each inventory may also be a consequence of the resolution of the optical satellite imagery or how easily visible small landslides were in optical imagery in each case, and not necessarily reflect real differences in the size distribution of landslides. In contrast, there is likely to be less sampling bias between different events for larger landslides and more intensely affected areas.
**Manuscript Change**
None

**Comment 34**
340 – 345 'The second method...': this sounds to me like an indirect evaluation because in between the real numbers and the results of the RF there is a further model to obtain the interpolation. If correct, fine for me but I suggest to better motivate the choice.
**Response**
High R2 values do not necessarily indicate a good model LAD, but low R2 values, such as those seen in this paper mean that the level of noise in the model is too high to obtain a useful model. Since the R2 values are so low anyway, we don't feel it is necessary to add in the gradient or intercept of the linear regression. Even if these were perfect, the low R2 value would mean our results would not change.
**Manuscript Change**
"*A high r2 coefficient value (up to a maximum of 1.0) indicates low levels of random errors and suggests that the model fits the observed LAD well and an r2 value of zero represents a model with no skill.*" at lines 343-344 of the original manuscript has been changed to "*A high r2 coefficient value (up to a maximum of 1.0) indicates low levels of random errors, while an r2 close to zero indicates*

*that levels of random errors in the model are too high to obtain a good fit to observed LAD*" at lines 346-347 of the revised manuscript.

**Comment 35**
3 results
3.2 Global models
General comment: I would have preferred to see the original ROCs and not only the differences.
**Response**
The ROC AUC values are visible on the figure and are also included in the text. At line 394 of the original manuscript, we had described only the change in ROC AUC. This has now been changed to include exact ROC AUC values.
**Manuscript Change**
We have changed "*the model consistently has an AUC of around 0.2 more than the individual Sentinel-1 coherence surfaces*" at line 394 of the original manuscript to
 "*the model has a consistently higher AUC than the individual Sentinel-1 coherence surfaces (approximately 0.8 compared to 0.6)*" at line 397-398 of the revised manuscript.

**Comment 36**
3.3 Do these models outperform individual InSAR coherence methods?
General comment: I'm not so sure that I got the point. Susceptibility and mapping are two different things and when you compare using the LAD benchmark you are comparing different results. In the first case the capacity of predicting spatial landslide occurrence (if this is a real susceptibility … what is more, including a sort of ground truth obtained from the coherence-based LAD), in the second you measure the capacity of mapping using a technique.
**Response**
InSAR coherence is not, in fact, a direct measurement of landslides and so cannot really be thought of as a ground truth. The ICFs used are indicative of changes to the ground surface, which include landslides but may also include changes to vegetation/ building damage/ ground surface rupture. The empirical model predicts where landslides might be expected to have happened following an earthquake of the observed characteristics. InSAR coherence methods and empirical models both provide imperfect information on the distribution of landslides immediately after an earthquake, and neither can be considered as ground-truth. We therefore think it is useful to compare these two different approaches to estimating the distribution of earthquake-triggered landslides (as measured from mapping of optical satellite imagery).
**Manuscript Change**
None

**Comment 37**
3.4 Feature importances
410 'where these become the most important feature in the model': Here there are at least 3 different 'categories' of features: (I) those features that take a static picture of the pre-event situation (lithology, slope…) (II) the velocity, which is strongly contingent upon the event, and (III) the Insar features, that map the event. An RF feature importance evaluation is only numerical, I suggest providing a geomorphological interpretation, and considering whether the 3 classes can be really evaluated together using the same criterion.
**Response**
This evaluation is only numerical, and it is limited in that there are dependencies between some of the categories (e.g. coherence is influenced by slope and land cover). However, it clearly illustrates the growing role for ICFs over time after the earthquake and that is the primary purpose of this analysis. Geomorphological interpretation is not the focus of the paper but we do take this

opportunity to clarify why these different data sources can be evaluated together using the same criterion.

**Manuscript Change**

Additional text to Section 3.4, Lines 415-417 of the revised manuscript
*"As described in Section 2.4.1, these measures of importance are limited since our input features are not fully independent, but it is clear that the importance of the ICFs increases through time in each case."*

**Comment 38**

4 Discussion

General comment: I in principle agree with all the topics chosen for the discussion, but I think that a real discussion on the geomorphological interpretation of the results is missing.

**Response**

We believe that a geomorphological interpretation, while of interest for future studies, is beyond the scope of this study, since we are considering methods of generating information for use in an emergency response context and not for use in understanding the changes caused by triggered landslides to the landscape.

**Manuscript Change** None

**Comment 39**

4.1 Model interpretation based on ROC AUC and r2

420 – 425 'how the current generation of empirical models at this spatial scale should be interpreted.?: any hint, personal opinion, on this?

**Response**

Our opinion was given in the following sentences, we have corrected this to make it clearer.

**Manuscript Change**

*"This result directly impacts how the current generation of empirical models at this spatial scale should be interpreted. The low r2 values we have observed indicate that the ability of the models to predict LAD as a continuous variable is poor. The more encouraging AUC values indicate that the models are well-suited to discriminating between affected and unaffected pixels."* at lines 422-425 of the original manuscript has been changed to

*"This result directly impacts how the current generation of empirical models at this spatial scale should be interpreted: the low r2 values we have observed indicate that the ability of the models to predict LAD as a continuous variable is poor, while the more encouraging AUC values indicate that the models are well-suited to discriminating between affected and unaffected pixels."* at Lines 433-436 of the revised manuscript.

**Comment 40**

4.5 Current recommendations for best practice

General comment on L-C bands: probably one of the things in common among the 3 events is the presence of vegetation in the affected areas before the event. Would your suggestion change if lc was different? See recent earthquakes in Iran that triggered landslides.

**Response**

It is true that our recommendations apply only to vegetated areas. We now specify this at the beginning of this section. We also already explicitly address the case of more sparsely vegetated areas in the discussion (Section 4.7 of the revised manuscript) – indeed this is an important area for further research

**Manuscript Change**

Added *"in vegetated areas"* at line 520 of the revised manuscript.

Added *"in a vegetated area"* at line 523 of the revised manuscript.

**Comment 41**

525 – 535: I reiterate a previous comment: can the two products be directly compared? One is mapping, the other is 'probabilistic spatial forecasting'. I agree with the suggestion about having a single product to update.

**Response**

We believe the comparison is useful since both products are developed for the same application and can be assessed using ROC analysis. (See response to comment 36)

**Comment 42**

545 'is thus not usually reliable in landslide detection...' I suggest relax this sentence, actually, it is, it just needs more controls that are not highlighted in the cited studies.

**Response**

Random Forests rely on establishing thresholds when splitting the data. Since landslides can cause both increases and decreases in amplitude and background amplitude is dependent on numerous factors including surface roughness, soil moisture content and particularly on slope orientation relative to sensor, we expect this method would struggle.

**Manuscript Change**

"*However, amplitude depends on several factors including soil moisture content and slope orientation relative to the satellite sensor. Landslides can therefore result in both increases and decreases in amplitude, and false positives can easily arise. Amplitude change is thus not usually reliable in landslide detection (Czuchlewski et al. 2003; Park and Lee, 2019)*" at lines 543-545 of the original manuscript changed to

"*However, other studies have found amplitude to be unreliable in landslide detection (Czuchlewski et al., 2003; Park and 555 Lee, 2019). SAR amplitude depends on several factors including surface roughness, soil moisture content and slope orientation relative to the satellite sensor. The amplitude of non-landslide pixels is therefore likely to vary significantly across the study area. Landslides can result in both increases and decreases in radar amplitude. Since the Random Forests technique relies on establishing thresholds on which to divide data, input features based on amplitude methods may not be suitable.*" at lines 554-558 of the revised manuscript.

**Comment 43**

560 – 565: See my previous comment on C-L bands, I suggest mentioning these parts earlier in the recommendations for best practices.

**Response**

We mention them here as a motivation for possibly using data from the planned NiSAR L-band mission in the future.

**Manuscript** None

**Comment 44**

4.7 Possible applications to rainfall-triggered landslides
General comment, which can be somehow applied to the next paragraph as well: it seems to me too generic, and I'm not so sure that this topic deserves a dedicated paragraph. I suggest shortening this part.
 4.8 Possible application in arid environments
 see my previous comment

**Response**

We will combine these two sections into a single section "application in other settings"

**Manuscript Change**

Sections 4.7 and 4.8 of the original manuscript have been combined to form Section 4.7 of the revised manuscript. The text is unchanged, other than the sentence "*It is also worth considering the*

*possible application of our approach in an arid environment.”* which was added at line 603 of the revised manuscript to link the two original sections together.

**Reviewer 2**
**Comment 1**
My main suggestion is to provide more information on the influence of wavelength (L-band) on the accuracy of the results. The difference between sentinel-1 and PALSAR-2 is not only the wavelength but also the resolution, polarization, and incidence angle. In particular, incidence angle or local incidence angle are important for landslide detection. I think that authors should consider these as well.
**Response**
Polarisation: ALOS-2 acquires quad-pol data while Sentinel-1 acquires dual-pol only (single-pol for events early in the lifetime of the satellite e.g. the 2015 Nepal earthquake). However, here we use only single-pol data from both satellites. This is now specified at line 230. We chose to do this as single-pol SAR data are more likely to be available immediately after an earthquake. Examining the influence of fully polarimetric SAR on empirical models is beyond the scope of this study but would be interesting for future work.
ALOS-2 data is acquired at a larger range of incidence angles (8°-70° degrees) compared to Sentinel-1 (29.1°-46.0°). However, all data used in this study were acquired within a relatively limited range of angles (31.4°-43.8°) so that the incidence angle should be fairly similar for Sentinel-1 and PALSAR-2. These incidence angles are now provided at line 237.
For information on the spatial and temporal resolutions of the two datasets, we direct readers to the paper from which these data were taken, where their processing is described in detail (Burrows et al. 2020).
**Manuscript Change**
“*We used SAR data*” at line 228 of the original manuscript changed to “*We used single-polarisation SAR data*” at line 229 of the revised manuscript
“*Here all data were acquired at an angle of 31.4°-43.8° to vertical.*” added at line 236-237 of the revised manuscript.
“*Further details on parameter choices made in the generation of CECL, Bx-S, PECI, ΔC_sum and ΔC_max surfaces can be found in Burrows et al. (2020).*” at lines 241-242 of the original manuscript changed to “*Further details on the spatial and temporal resolution of these SAR data, on their processing and on parameter choices made in the generation of the CECL, Bx-S, PECI, ΔC_sum and ΔC_max surfaces can be found in Burrows et al. (2020).*” at lines 243-244 of the revised manuscript.

**Comment 2**
Furthermore, I have a question in assessing model performance. Authors use ROC analysis based on the Burrows et al. (2020). However, ROC analysis requires the creation of binary landslide images. The binarization of landslide areal density (LAD) only degrades the image. I think R2 is fair for assessing the LAD prediction. Authors should describe the effectiveness of ROC analysis.
**Response**
We use ROC analysis as it is a metric commonly used in landslide detection / prediction studies and is easy to understand. Although this requires us to convert the landslide density to a binary surface, we do not suggest using the binary surface as the final product, as you are right this would lead to a loss of information. ROC AUC as we use it here provides an indication of the skill with which areas of particularly intense landsliding can be identified.
By presenting both ROC AUC values and R2 values, we feel a more complete picture of model performance is obtained than by using either one of these metrics alone. Indeed, Reviewer 1 indicates in their review that they prefer the use of ROC as a metric over the use of R2 (Reviewer 1 Comment 34), so we believe the presentation of both metrics represents a good balance.
**Manuscript Change** None

**Comment 3**

698 – 699 I think that Masato is given name. Please correct it.

**Response**

We apologise for this oversight, this has been corrected in the references and in the main document.

**Manuscript Change**

In text changes from "*Masato et al. 2020*" to "*Ohki et al. 2020*" at lines 53, 56, 127, 132, 292, 562

Change to reference list at line 723 of the revised manuscript

Previously :

*Masato, O., Abe, T., Takeo, T., and Masanobu, S.: Landslide detection in mountainous forest areas using polarimetry and interferometric coherence, Earth, Planets and Space (Online), 72, 2020.*

Now:

*Ohki, M., Takahiro, A., Tadono, T., and Shimada, M.: Landslide detection in mountainous forest areas using polarimetry and interferometric coherence, Earth, Planets and Space (Online), 72, 2020.*

**Additional comments from Editor**

**Comment 1**

Following a suggestion by referee #1 you have proposed to modify the title in "Integrating empirical models and satellite radar improves landslide prediction for emergency response". I suggest that perhaps a better choice would be "Integrating empirical models and satellite radar can improve landslide detection for emergency response". I understand the concern of referee #1 about "detection" vs. "prediction" when using empirical models, but I in my opinion "detection" is more suitable, as you aim to reproduce/reconstruct something that has already occurred, and not something that has to occur ("reconstruction" is perhaps also a suitable term). Also "can" should be used, as more studies should perhaps further corroborate your results

**Response**

Thank you for the suggestion, we agree that this title best describes the manuscript.

**Manuscript Change**

Manuscript title changed from "*Improved rapid landslide detection from integration of empirical models and satellite radar*" to "*Integrating empirical models and satellite radar can improve landslide detection for emergency response*"

**Comment 2**

In the "global model" you use three events, which is a low number. I was wondering if you have tried to apply a "leave one out" cross-validation (See e.g. https://link.springer.com/referenceworkentry/10.1007%2F978-0-387-30164-8_469) where "one" is intended as "one earthquake event".

**Response**

We had applied leave-one-out cross-validation in Section 3.2, however we will alter the text and include the name of this technique to improve clarity.

**Manuscript Change**

"*To test the effect that incorporating ICFs might have on such models using our three case studies, we trained models on two earthquakes and predicted LAD triggered by the third.*" at line 368-369 of the original manuscript changed to "*To test the effect that incorporating ICFs might have on such models using our three case studies, we applied a "leave-one-out cross-validation" approach: we tested predictions of LAD for each earthquake that were generated using models trained on the other two earthquakes.*" at lines 371-372 of the revised manuscript.